# Towards Symmetry-Aware Generation of Periodic Materials

**Youzhi Luo, Chengkai Liu, Shuiwang Ji**
Department of Computer Science & Engineering
Texas A&M University
College Station, TX 77843
{yzluo,liuchengkai,sji}@tamu.edu

## Abstract

We consider the problem of generating periodic materials with deep models. While symmetry-aware molecule generation has been studied extensively, periodic materials possess different symmetries, which have not been completely captured by existing methods. In this work, we propose SyMat, a novel material generation approach that can capture physical symmetries of periodic material structures. SyMat generates atom types and lattices of materials through generating atom type sets, lattice lengths and lattice angles with a variational auto-encoder model. In addition, SyMat employs a score-based diffusion model to generate atom coordinates of materials, in which a novel symmetry-aware probabilistic model is used in the coordinate diffusion process. We show that SyMat is theoretically invariant to all symmetry transformations on materials and demonstrate that SyMat achieves promising performance on random generation and property optimization tasks. Our code is publicly available as part of the AIRS library (https://github.com/divelab/AIRS).

## 1   Introduction

Designing or synthesizing novel periodic materials with target properties is a fundamental problem in many real-world applications, such as designing new periodic materials for solar cells and batteries [1]. For a long time, this challenging task heavily relies on either manually designing material structures based on the experience of chemical experts, or running very expensive and time-consuming density functional theory (DFT) based simulation. Recently, thanks to the progress of deep learning techniques, many studies have applied advanced deep generative models [22, 13] to generate or discover novel chemical compounds, such as molecules [10, 39, 50] and proteins [48, 16]. However, while deep learning has been widely used in periodic material representation learning [51, 5, 40, 4, 57, 25], generating novel periodic materials with deep generative models remains largely under-explored.

The major challenge of employing deep generative models to the generation of periodic materials is capturing physical symmetries of periodic material structures. Ideally, deep generative models for periodic materials should maintain invariance to symmetry transformations of periodic materials, including permutation, rotation, translation, and periodic transformations [59, 9, 60]. To achieve this target, we propose SyMat, a novel symmetry-aware periodic material generation method. SyMat transforms the atom types and lattices of materials to symmetry-aware generation targets, which are generated by a variational auto-encoder model [22]. Besides, SyMat adopts novel symmetry-aware probabilistic diffusion process to generate atom coordinates with a score-based diffusion model [44]. Experiments show that SyMat can achieve promising performance on random generation and property optimization tasks.

37th Conference on Neural Information Processing Systems (NeurIPS 2023).

**Relations with Prior Methods.** We note that several existing studies are related to our proposed SyMat, such as CDVAE [52]. Generally, the major difference of SyMat between them lies in the capture of symmetries. We will discuss the detailed differences in Section 3.5.

## 2 Background and Related Work

### 2.1 Molecule Generation

Designing or discovering novel molecules with target properties has long been a challenging problem in many chemical applications. Recently, motivated from the significant progress in deep generative learning [22, 12, 44, 13], a lot of studies have applied advanced deep generative models to this problem. Some early studies [19, 58, 41, 32, 27] consider molecules as 2D topology graphs and design graph generation models to discover novel molecules. More recently, considering the importance of 3D structures in many molecular properties, many studies have also proposed advanced generative models that can generate 3D molecular conformations either from scratch [10, 39, 31, 15, 50], or from input conditional information [33, 56, 43, 53, 54, 42, 55, 20, 28]. However, all these existing studies are designed for non-periodic molecules while materials have periodic structures, in which atoms periodically repeat themselves in 3D space (see Figure 1 for an illustration of differences between molecules and periodic materials). Hence, they cannot be directly applied to the periodic material generation problem that we study in this work.

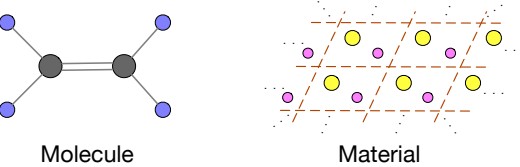

Molecule      Material

Figure 1: An illustration of molecule (left) and periodic material (right) structures. Different from molecules, atoms in materials periodically repeat themselves infinitely in 3D space (we use a 2D visualization here for simplicity).

### 2.2 Periodic Material Generation

While molecule generation has been extensively studied, currently the periodic material design or generation problem remains largely under-explored. Early periodic material design methods [34, 7] only generate compositions of chemical elements in periodic materials but do not generate 3D structures. More recent studies have proposed to use variational auto-encoders (VAEs) [22] or generative adversarial networks (GANs) [12] to generate 3D structures of periodic materials. However, these methods use coordinate matrices [36, 21, 61] or voxel images [14, 6] of materials as the direct generation targets, which results in the violation of physical symmetries of periodic material structures in their captured probabilistic distributions. Also, the VAE and GAN models used by them are not powerful enough to capture the distribution of complicated 3D periodic material structures. Different from them, our method employs a more powerful score-based diffusion model [44] to generate 3D periodic material structures, and the model is designed to capture physical symmetries in materials.

### 2.3 Score-Based Diffusion Models for 3D Structure Generation

Diffusion models are a family of deep generative models that have achieved outstanding generation performance in a variety of data modalities, such as images [38, 37], audios [24], and molecular conformations [55, 15, 20]. Currently, score-based diffusion models [44] and denoising diffusion probabilistic models [13] are two most commonly used diffusion models, and our method employs score-based diffusion models for 3D periodic material generation. For a given data distribution $p(\boldsymbol{x})$ in the high-dimensional data space, the score-based diffusion model trains a parameterized score model $\boldsymbol{s}_\theta(\cdot)$ with parameters $\theta$ to accurately approximate the score function $\nabla_{\boldsymbol{x}} \log p(\boldsymbol{x})$ from $\boldsymbol{x}$. To improve the score approximation accuracy in the regions where the training data is sparse, denoising score matching [46, 44] is proposed to more effectively train $\boldsymbol{s}_\theta(\cdot)$. Specifically, let $\{\sigma_t\}_{t=1}^T$ be a set of positive scalars satisfying $\sigma_1 > \sigma_2 > ... > \sigma_T$, the denoising score matching method first perturbs the data point $\boldsymbol{x}$ sampled from $p(\boldsymbol{x})$ by a sequence of Gaussian noises with noise magnitude levels $\{\sigma_t\}_{t=1}^T$, *i.e.*, sampling $\tilde{\boldsymbol{x}}$ from $p_{\sigma_t}(\tilde{\boldsymbol{x}}|\boldsymbol{x}) = \mathcal{N}(\tilde{\boldsymbol{x}}|\boldsymbol{x}, \sigma_t^2 \boldsymbol{I})$ for every noise level $\sigma_t$. The score model $\boldsymbol{s}_\theta(\cdot)$ is optimized to predict the denoising score function $\nabla_{\tilde{\boldsymbol{x}}} \log p_{\sigma_t}(\tilde{\boldsymbol{x}}|\boldsymbol{x})$ from $\tilde{\boldsymbol{x}}$ at each noise level, where the ground truth of $\nabla_{\tilde{\boldsymbol{x}}} \log p_{\sigma_t}(\tilde{\boldsymbol{x}}|\boldsymbol{x})$ can be easily computed from the mathematical formula

of $\mathcal{N}(\tilde{\boldsymbol{x}}|\boldsymbol{x}, \sigma_t^2 \boldsymbol{I})$. Formally, $\boldsymbol{s}_\theta(\cdot)$ is optimized to minimize the following training objective $\mathcal{L}$:

$$\mathcal{L} = \frac{1}{2T} \sum_{t=1}^{T} \sigma_t^2 \mathbb{E}_{\tilde{\boldsymbol{x}}, \boldsymbol{x}} \left[ \left\| \frac{\boldsymbol{s}_\theta(\tilde{\boldsymbol{x}})}{\sigma_t} + \frac{\tilde{\boldsymbol{x}} - \boldsymbol{x}}{\sigma_t^2} \right\|_2^2 \right]. \tag{1}$$

As shown in Vincent [46], when $\sigma_t$ is small enough, after $\boldsymbol{s}_\theta(\cdot)$ is well trained with the objective in Equation (1), the output $\boldsymbol{s}_\theta(\boldsymbol{x})$ can accurately approximate the exact score function for any input $\boldsymbol{x}$. New data samples can be generated by running multiple steps of the Langevin dynamics sampling algorithm [49] with the score functions approximated by the score model.

Several existing studies have applied score-based diffusion models to 3D molecular and material structure generation. ConfGF [42] is the first score-based diffusion model for 3D molecular conformation generation. It proposes a novel theoretic framework to achieve roto-translational equivariance in score functions. Based on the theoretic framework of ConfGF, DGSM [30] further improves the performance of ConfGF by dynamic graph score matching. Though they have achieved good performance in 3D molecules, they do not consider periodic invariance so they cannot be applied to 3D periodic materials. Recently, a novel approach CDVAE [52] is proposed for periodic material generation, and score-based diffusion models are used in CDVAE to generate atom coordinates in materials. However, as we discussed in Section 3.5, CDVAE fails to achieve translational invariance in calculating the denoising score function, so it does not capture all physical symmetries.

## 3 SyMat: Symmetry-Aware Generation of Periodic Materials

While a variety of deep generative models have been proposed to capture all physical symmetries in 3D molecular conformations, it remains challenging to do it for periodic materials as they have more complicated 3D periodic structures. In this section, we present SyMat, a novel periodic material generation method. In the following subsections, we will first introduce the problem of symmetry-aware material generation, then elaborate how our proposed SyMat approach achieves invariance to all symmetry transformations of periodic materials.

### 3.1 Symmetry-Aware Generation

For any periodic material structure, we can consider it as periodic repetitions of one unit cell in 3D space, where any unit cell is the smallest repeatable structure of the material. Hence, for any periodic material $\boldsymbol{M}$, we can describe its complete structure information with one of its unit cells and the three lattice vectors describing its periodic repeating directions. Specifically, assuming there are $n$ atoms in any unit cell of $\boldsymbol{M}$, we represent $\boldsymbol{M}$ as $\boldsymbol{M} = (\boldsymbol{A}, \boldsymbol{P}, \boldsymbol{L})$, where $\boldsymbol{A} \in \mathbb{Z}^n$, $\boldsymbol{P} \in \mathbb{R}^{3 \times n}$, and $\boldsymbol{L} \in \mathbb{R}^{3 \times 3}$ are the atom type vector, coordinate matrix, and lattice matrix, respectively. Here, the $i$-th element $a_i$ of $\boldsymbol{A} = [a_1, ..., a_n]$ and the $i$-th column vector $\boldsymbol{p}_i$ of $\boldsymbol{P} = [\boldsymbol{p}_1, ..., \boldsymbol{p}_n]$ denote the atom type, *i.e.*, atomic number and the 3D Cartesian coordinate of the $i$-th atom in the unit cell, respectively. The lattice matrix $\boldsymbol{L} = [\boldsymbol{\ell}_1, \boldsymbol{\ell}_2, \boldsymbol{\ell}_3]$ is formed by three lattice vectors $\boldsymbol{\ell}_1, \boldsymbol{\ell}_2, \boldsymbol{\ell}_3$ indicating how the atoms in a unit cell periodically repeat themselves in 3D space.

In this work, we consider the problem of generating periodic material structures with generative models. Formally, we are given a dataset $\mathcal{M} = \{\boldsymbol{M}_j\}_{j=1}^m$ where each periodic material data $\boldsymbol{M}_j$ is assumed to be sampled from the distribution $p(\cdot)$. Because of the physical symmetry properties in periodic materials [59, 9], $p(\cdot)$ is restricted to be invariant to the following symmetry transformations.

- **Permutation.** For any $\boldsymbol{M} = (\boldsymbol{A}, \boldsymbol{P}, \boldsymbol{L})$, if we permute $\boldsymbol{A}$, *i.e.*, exchange elements in $\boldsymbol{A}$ to obtain $\boldsymbol{A}'$, and use the same permutation order to permute the column vectors of $\boldsymbol{P}$ to obtain $\boldsymbol{P}'$, we actually change nothing but atom orders in unit cell representations. Hence, let $\boldsymbol{M}' = (\boldsymbol{A}', \boldsymbol{P}', \boldsymbol{L})$, the probability densities of $\boldsymbol{M}$ and $\boldsymbol{M}'$ are actually the same, *i.e.*, $p(\boldsymbol{M}) = p(\boldsymbol{M}')$.

- **Rotation and translation.** For any $\boldsymbol{M} = (\boldsymbol{A}, \boldsymbol{P}, \boldsymbol{L})$, we can rotate and translate its atom positions in 3D space to obtain a new material $\boldsymbol{M}' = (\boldsymbol{A}, \boldsymbol{P}', \boldsymbol{L}')$, where $\boldsymbol{P}' = \boldsymbol{Q}\boldsymbol{P} + \boldsymbol{b}\mathbf{1}^T$ and $\boldsymbol{L}' = \boldsymbol{Q}\boldsymbol{L}$. Here $\boldsymbol{Q} \in \mathbb{R}^{3 \times 3}$ is the rotation matrix satisfying $\boldsymbol{Q}^T\boldsymbol{Q} = \boldsymbol{I}$, $\boldsymbol{b} \in \mathbb{R}^3$ is the translation vector, and $\mathbf{1}$ is an $n$-dimensional vector whose elements are all 1. It can be intuitively understood that rotation and translation do not change any internal structures of periodic materials, so $\boldsymbol{M}$ and $\boldsymbol{M}'$ are actually different representations of the same material, so $p(\boldsymbol{M}) = p(\boldsymbol{M}')$ should hold.

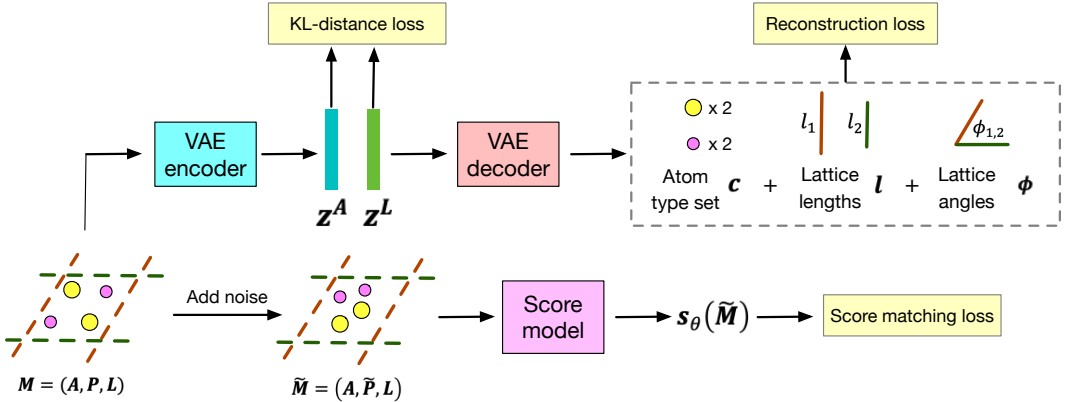

(a) The overall process of training the SyMat model on a data sample $M = (A, P, L)$ from the dataset.

(b) The overall process of generating a new material with the SyMat model.

Figure 2: Illustrations of training and generation process in our proposed SyMat method.

- **Periodic transformation.** Because every atom from a unit cell periodically repeats itself along lattice vectors in 3D space, any coordinate matrix $P \in \mathbb{R}^{3 \times n}$ has infinite number of periodically equivalent coordinate matrices. Formally, a periodically equivalent coordinate matrix $P^K$ of $P$ can be obtained by periodic transformation as $P^K = P + LK$, where $K \in \mathbb{Z}^{3 \times n}$ is an arbitrary integer matrix. Since $P^K$ and $P$ are equivalent and the periodic transformations between them are invertible, we consider $M = (A, P, L)$ and $M^K = (A, P^K, L)$ as the same material so $p(M) = p(M^K)$ should hold.

In the periodic material generation problem, our target is to learn a generative model $p_\theta(\cdot)$ with parameters $\theta$ from the given dataset $\mathcal{M}$ so that the model can capture the real data distribution $p(\cdot)$, and the learned generation model can generate a valid material structure $M$ with a high probability $p_\theta(M)$. In addition, we require the distribution $p_\theta(\cdot)$ captured by the generation model to have the same symmetry-related restrictions as the real data distribution $p(\cdot)$. The main motivation of modeling symmetries comes from property optimization. A significant target of developing generative models for periodic materials is to generate novel periodic materials with desirable chemical properties (*e.g.*, low energies). Notably, chemical properties are invariant to physical symmetry transformations. For instance, rotating a material structure in 3D space does not change its energy. In other words, chemical properties only depend on the 3D geometric information that is invariant to symmetry transformations (*e.g.*, interatomic distances). Using symmetry-aware modeling forces generative models to only capture the distribution of these 3D geometric information, which facilitates searching materials with desirable properties in the latent space.

### 3.2 An Overview of SyMat

Generally, to generate a new material $M = (A, P, L)$, our proposed SyMat method first generates the atom type vector $A$ and lattice matrix $L$, then generates the coordinate matrix $P$ conditioned on $A$ and $L$. The probability of generating $M = (A, P, L)$ can then be described as

$$p_\theta(M) = p_\theta(A)p_\theta(L)p_\theta(P|A, L). \tag{2}$$

Since the structures of atom types and lattices are relatively simple, we find that a VAE [22] model already has adequate capability to capture their distributions. Hence, we use a VAE model to generate $A$ and $L$. However, the structures of coordinates are much more complicated, so we use a powerful score-based diffusion model [44] for their generation. Next, we will elaborate the details of generating atom types and lattices in Section 3.3 and the details of generating coordinates in Section 3.4. An overview of the proposed SyMat method is illustrated in Figure 2.

## 3.3 Atom Type and Lattice Generation

For the generation of atom types and lattices, a natural idea is using the VAE model to directly generate and capture the distribution of $\boldsymbol{A}$ and $\boldsymbol{L}$. However, this fails to capture the invariance to symmetry transformations described in Section 3.1, including the permutation invariance of $\boldsymbol{A}$ and rotation invariance of $\boldsymbol{L}$. To tackle this issue, we transform $\boldsymbol{A}$ and $\boldsymbol{L}$ to items that are invariant to symmetry transformations, and make these items be the direct generation targets of the VAE model. Specifically, for $\boldsymbol{A} = [a_1, ..., a_n]$, we can count the number of atoms for every atom type existed in $\boldsymbol{A}$, and represent $\boldsymbol{A}$ with an unordered $k$-element atom type set $\boldsymbol{c} = \{(e_1, n_1), ..., (e_k, n_k)\}$, which indicates that for $i = 1, ..., k$, there exist $n_i$ atoms with the atom type $e_i$ in $\boldsymbol{A}$. We can easily find that $\boldsymbol{c}$ is invariant to any permutations on $\boldsymbol{A}$ so we use $\boldsymbol{c}$ as the direct generation targets. Besides, for the lattice matrix $\boldsymbol{L} = [\boldsymbol{\ell}_1, \boldsymbol{\ell}_2, \boldsymbol{\ell}_3]$, we do not generate the three lattice vectors $\boldsymbol{\ell}_1, \boldsymbol{\ell}_2, \boldsymbol{\ell}_3$ directly. Instead, we use six rotation-invariant items as the generation targets, including three lattice lengths $\boldsymbol{\ell} = [\ell_1, \ell_2, \ell_2]$, where $\ell_i = ||\boldsymbol{\ell}_i||_2$, and three lattice angles $\boldsymbol{\phi} = [\phi_{12}, \phi_{13}, \phi_{23}]$, where $\phi_{ij}$ is the angle between $\boldsymbol{\ell}_i$ and $\boldsymbol{\ell}_j$. Overall, we transform $\boldsymbol{A}$ and $\boldsymbol{L}$ to generation targets $\boldsymbol{c}, \boldsymbol{\ell}, \boldsymbol{\phi}$ so that the generation process and captured distributions $p_\theta(\boldsymbol{A})$, $p_\theta(\boldsymbol{L})$ are ensured to be symmetry-aware.

The VAE model used for generating $\boldsymbol{c}$, $\boldsymbol{\ell}$, and $\boldsymbol{\phi}$ consists of an encoder model and a decoder model. The encoder model is based on a 3D graph neural network (GNN) that takes a material $\boldsymbol{M} = (\boldsymbol{A}, \boldsymbol{P}, \boldsymbol{L})$ as inputs and outputs a $d$-dimensional latent variable $\boldsymbol{z}^{\boldsymbol{A}} \in \mathbb{R}^d$ and an $f$-dimensional latent variable $\boldsymbol{z}^{\boldsymbol{L}} \in \mathbb{R}^f$. This 3D GNN employs the symmetry-aware SphereNet [29] architecture so $\boldsymbol{z}^{\boldsymbol{A}}$, $\boldsymbol{z}^{\boldsymbol{L}}$ will not change if applying any transformations described in Section 3.1 to the input material. We will introduce more details of this 3D GNN model in Appendix B. The decoder model is composed of four multi-layer perceptrons (MLPs) including $\text{MLP}^e$, $\text{MLP}^k$, $\text{MLP}^n$, $\text{MLP}^{\boldsymbol{L}}$, which are used to predict $\boldsymbol{c} = \{(e_1, n_1), ..., (e_k, n_k)\}$, $\boldsymbol{\ell} = [\ell_1, \ell_2, \ell_3]$, and $\boldsymbol{\phi} = [\phi_{12}, \phi_{13}, \phi_{23}]$ from the latent variables $\boldsymbol{z}^{\boldsymbol{A}}$ and $\boldsymbol{z}^{\boldsymbol{L}}$. Specifically, let $E$ be the number of all considered atom types and $N$ be the largest possible number of atoms that can exist in a material, $\text{MLP}^e$ predicts a vector $\boldsymbol{p}_e \in [0, 1]^E$ which contains the existing probability for each of the $E$ atom type from the input $\boldsymbol{z}^{\boldsymbol{A}}$, and $\text{MLP}^k$ predicts the size $k$ of the set $\boldsymbol{c}$ from the input $\boldsymbol{z}^{\boldsymbol{A}}$. The $k$ atom types $e_1, ..., e_k$ with top-$k$ probabilities in $\boldsymbol{p}_e$ are chosen as the atom types in $\boldsymbol{c}$, and $\text{MLP}^n$ predicts the number of atoms $n_1, ..., n_k$ for them, where each $n_i$ is predicted from $e_i$ and $\boldsymbol{z}^{\boldsymbol{A}}$. Here, the prediction of $\boldsymbol{p}_e$ is considered as $E$ binary classification tasks for each of $E$ atom types, and the prediction of $k$ and $n_1, ..., n_k$ are both considered as $N$-category classification tasks. Besides, $\text{MLP}^{\boldsymbol{L}}$ is used to predict lattice items $[\boldsymbol{\ell}, \boldsymbol{\phi}] = [\ell_1, \ell_2, \ell_3, \phi_{12}, \phi_{13}, \phi_{23}]$ from $\boldsymbol{z}^{\boldsymbol{L}}$, which is considered as a regression task.

We now describe the training and generation process of the VAE model. When training the VAE model on a material dataset $\mathcal{M}$, for any material $\boldsymbol{M} = (\boldsymbol{A}, \boldsymbol{P}, \boldsymbol{L})$ in $\mathcal{M}$, the encoder model first maps it to latent variable $\boldsymbol{z}^{\boldsymbol{A}}$ and $\boldsymbol{z}^{\boldsymbol{L}}$. Afterwards, the decoder model uses $\boldsymbol{z}^{\boldsymbol{A}}$ and $\boldsymbol{z}^{\boldsymbol{L}}$ to reconstruct the exact $\boldsymbol{c}, \boldsymbol{\ell}$, and $\boldsymbol{\phi}$ obtained from $\boldsymbol{A}$ and $\boldsymbol{L}$. The VAE model is optimized to minimize the reconstruction error of $\boldsymbol{c}, \boldsymbol{\ell}, \boldsymbol{\phi}$ together with the KL-distance between $\boldsymbol{z}^{\boldsymbol{A}}$, $\boldsymbol{z}^{\boldsymbol{L}}$ and the standard Gaussian distribution $\mathcal{N}(\boldsymbol{0}, \boldsymbol{I})$. The reconstruction error losses are defined as cross entropy loss and minimum squared error loss for classification and regression tasks, respectively. After the model is trained, we can generate $\boldsymbol{A}$ and $\boldsymbol{L}$ by first sampling latent variables $\boldsymbol{z}^{\boldsymbol{A}}$, $\boldsymbol{z}^{\boldsymbol{L}}$ from $\mathcal{N}(\boldsymbol{0}, \boldsymbol{I})$, then using the decoder to map $\boldsymbol{z}^{\boldsymbol{A}}$, $\boldsymbol{z}^{\boldsymbol{L}}$ to $\boldsymbol{c}, \boldsymbol{\ell}, \boldsymbol{\phi}$. Finally, we can produce $\boldsymbol{A}$ from $\boldsymbol{c} = \{(e_1, n_1), ..., (e_k, n_k)\}$ as

$$\boldsymbol{A} = [\underbrace{e_1, ..., e_1}_{n_1}, \underbrace{e_2, ..., e_2}_{n_2}, ..., \underbrace{e_k, ..., e_k}_{n_k}],$$

and produce $\boldsymbol{L}$ from $\boldsymbol{\ell}, \boldsymbol{\phi}$ by constructing three lattice vectors with lengths in $\boldsymbol{\ell}$ and pairwise angles in $\boldsymbol{\phi}$. The detailed procedure of producing three lattice vectors is described in Appendix B.

## 3.4 Coordinate Generation

After the atom type vector $\boldsymbol{A}$ and lattice matrix $\boldsymbol{L}$ are generated, our method will generate the coordinate matrix $\boldsymbol{P}$ conditioned on $\boldsymbol{A}$ and $\boldsymbol{L}$. We use a score-based diffusion model for coordinate generation. However, because the original score-based diffusion model proposed by Song and Ermon [44] does not consider any invariance to symmetry transformations in data distributions, we cannot directly apply its theoretic framework to coordinate generation. To overcome this limitation, we propose a novel probabilistic modeling process to calculate the score function of $\boldsymbol{P}$, *i.e.*, the score matrix $\nabla_{\boldsymbol{P}} \log p(\boldsymbol{P}|\boldsymbol{A}, \boldsymbol{L})$, and design the architecture and training framework of the score model to

achieve invariance to all symmetry transformations of crystal materials. We will elaborate the details of our proposed coordinate generation framework in the next paragraphs.

We first elaborate the restrictions of the score matrix $\nabla_{\boldsymbol{P}} \log p(\boldsymbol{P}|\boldsymbol{A}, \boldsymbol{L})$ that is supposed to be captured by the score model. By the following Proposition 3.1 (see its proof in Appendix A.1), we can show that if the material distribution $p(\boldsymbol{M})$ is invariant to all symmetry transformations described in Section 3.1, the score matrix $\nabla_{\boldsymbol{P}} \log p(\boldsymbol{P}|\boldsymbol{A}, \boldsymbol{L})$ will be invariant to translation and periodic transformations, and equivariant to permutation and rotation transformations.

**Proposition 3.1.** *If the material distribution $p(\boldsymbol{M})$ is invariant to permutation, rotation, translation, and periodic transformations described in Section 3.1, $\nabla_{\boldsymbol{P}} \log p(\boldsymbol{P}|\boldsymbol{A}, \boldsymbol{L})$ will satisfy the following properties: (1) $\nabla_{\boldsymbol{P}} \log p(\boldsymbol{P}|\boldsymbol{A}, \boldsymbol{L})$ is equivariant to permutations on $\boldsymbol{A}$ and $\boldsymbol{P}$; (2) $\nabla_{\boldsymbol{Q}\boldsymbol{P}+\boldsymbol{b}\boldsymbol{1}^T} \log p(\boldsymbol{Q}\boldsymbol{P} + \boldsymbol{b}\boldsymbol{1}^T|\boldsymbol{A}, \boldsymbol{Q}\boldsymbol{L}) = \boldsymbol{Q}\nabla_{\boldsymbol{P}} \log p(\boldsymbol{P}|\boldsymbol{A}, \boldsymbol{L})$ holds for any $\boldsymbol{Q} \in \mathbb{R}^{3 \times 3}, \boldsymbol{Q}^T\boldsymbol{Q} = \boldsymbol{I}$ and $\boldsymbol{b} \in \mathbb{R}^3$; (3) $\nabla_{\boldsymbol{P}+\boldsymbol{L}\boldsymbol{K}} \log p(\boldsymbol{P} + \boldsymbol{L}\boldsymbol{K}|\boldsymbol{A}, \boldsymbol{L}) = \nabla_{\boldsymbol{P}} \log p(\boldsymbol{P}|\boldsymbol{A}, \boldsymbol{L})$ holds for any $\boldsymbol{K} \in \mathbb{Z}^{3 \times n}$.*

To satisfy all properties in Proposition 3.1, we formulate $\nabla_{\boldsymbol{P}} \log p(\boldsymbol{P}|\boldsymbol{A}, \boldsymbol{L})$ as a function of score functions of edge distances in a graph created by multi-graph method [51]. For a periodic material $\boldsymbol{M}$ with coordinate matrix $\boldsymbol{P} = [\boldsymbol{p}_1, ..., \boldsymbol{p}_n]$ and lattice matrix $\boldsymbol{L}$, multi-graph method produces an $n$-node undirected graph on $\boldsymbol{M}$. In this graph, the node $i$ corresponds to the $i$-th atom in the unit cell, and an edge is added between any two nodes $i, j$ if one of the interatomic distances between them is smaller than a pre-defined cutoff $r$. Here, the interatomic distances between $i, j$ are the distances between their corresponding atoms in the complete infinite structure of $\boldsymbol{M}$, including both the distance within the unit cell and the distances crossing the unit cell boundary. Formally, the set of all edges in the graph constructed on $\boldsymbol{M}$ can be written as $\boldsymbol{E}(\boldsymbol{M}) = \{(i, j, \boldsymbol{k}) : ||\boldsymbol{p}_i + \boldsymbol{L}\boldsymbol{k} - \boldsymbol{p}_j||_2 \leq r, \boldsymbol{k} \in \mathbb{Z}^3, 1 \leq i, j \leq n\}$. Note that there may exist more than one edges connecting the nodes $i, j$ if more than one interatomic distances between them is smaller than $r$. Let $d_{i,j,\boldsymbol{k}} = ||\boldsymbol{p}_i + \boldsymbol{L}\boldsymbol{k} - \boldsymbol{p}_j||_2$ be the distance of the edge $(i, j, \boldsymbol{k})$, we consider $\log p(\boldsymbol{P}|\boldsymbol{A}, \boldsymbol{L})$ as a function of the distances of all edges in $\boldsymbol{E}(\boldsymbol{M})$. Denoting $\boldsymbol{s}_i$ as the score function of $\boldsymbol{p}_i$, then $\nabla_{\boldsymbol{P}} \log p(\boldsymbol{P}|\boldsymbol{A}, \boldsymbol{L}) = [\boldsymbol{s}_1, ..., \boldsymbol{s}_n]$. From the chain rule of derivatives, we can calculate $\boldsymbol{s}_i$ as

$$\boldsymbol{s}_i = \sum_{(j,\boldsymbol{k}) \in \mathcal{N}(i)} \nabla_{d_{i,j,\boldsymbol{k}}} \log p(\boldsymbol{P}|\boldsymbol{A}, \boldsymbol{L}) \cdot \nabla_{\boldsymbol{p}_i} d_{i,j,\boldsymbol{k}} = \sum_{(j,\boldsymbol{k}) \in \mathcal{N}(i)} s_{i,j,\boldsymbol{k}} \cdot \frac{\boldsymbol{p}_i + \boldsymbol{L}\boldsymbol{k} - \boldsymbol{p}_j}{d_{i,j,\boldsymbol{k}}}, \quad (3)$$

where the scalar $s_{i,j,\boldsymbol{k}} = \nabla_{d_{i,j,\boldsymbol{k}}} \log p(\boldsymbol{P}|\boldsymbol{A}, \boldsymbol{L})$ is the score function of the distance $d_{i,j,\boldsymbol{k}}$ and $\mathcal{N}(i) = \{(j, \boldsymbol{k}) : (i, j, \boldsymbol{k}) \in \boldsymbol{E}(\boldsymbol{M})\}$. With this probabilistic modeling process, $\nabla_{\boldsymbol{P}} \log p(\boldsymbol{P}|\boldsymbol{A}, \boldsymbol{L})$ can be approximated by first approximating $s_{i,j,\boldsymbol{k}}$ for every edge in the multi-graph representation of $\boldsymbol{M}$, then calculating each column vector of $\nabla_{\boldsymbol{P}} \log p(\boldsymbol{P}|\boldsymbol{A}, \boldsymbol{L})$ by Equation (3). By the following Proposition 3.2 (see its proof in Appendix A.2), we can show that if $s_{i,j,\boldsymbol{k}}$ is invariant to all symmetry transformations described in Section 3.1 for every edge $(i, j, \boldsymbol{k})$, the score matrix $\nabla_{\boldsymbol{P}} \log p(\boldsymbol{P}|\boldsymbol{A}, \boldsymbol{L})$ will satisfy all properties in Proposition 3.1.

**Proposition 3.2.** *For $\nabla_{\boldsymbol{P}} \log p(\boldsymbol{P}|\boldsymbol{A}, \boldsymbol{L}) = [\boldsymbol{s}_1, ..., \boldsymbol{s}_n]$ where each $\boldsymbol{s}_i$ is calculated by Equation (3) on a multi-graph representation of $\boldsymbol{M} = (\boldsymbol{A}, \boldsymbol{P}, \boldsymbol{L})$, if $s_{i,j,\boldsymbol{k}}$ is invariant to permutation, rotation, translation, and periodic transformations described in Section 3.1 for any edge $(i, j, \boldsymbol{k})$, then the calculated $\nabla_{\boldsymbol{P}} \log p(\boldsymbol{P}|\boldsymbol{A}, \boldsymbol{L})$ will always satisfy the properties in Proposition 3.1.*

In our method, we use a 3D GNN model as the score model $\boldsymbol{s}_\theta(\cdot)$ to approximate $\nabla_{\boldsymbol{P}} \log p(\boldsymbol{P}|\boldsymbol{A}, \boldsymbol{L})$. $\boldsymbol{s}_\theta(\cdot)$ takes the multi-graph representation of $\boldsymbol{M}$ as input and outputs $o_{i,j,\boldsymbol{k}}$ as the approximated $s_{i,j,\boldsymbol{k}}$ at every edge $(i, j, \boldsymbol{k})$ in the input graph. We use SphereNet [29] as the backbone architecture of $\boldsymbol{s}_\theta(\cdot)$, which ensures that the output $o_{i,j,\boldsymbol{k}}$ is always invariant to any symmetry transformation. More details about SphereNet will be introduced in Appendix B. To train the score model, we follow the most common operations in denoising score matching framework [44]. Specifically, for a material $\boldsymbol{M} = (\boldsymbol{A}, \boldsymbol{P}, \boldsymbol{L})$ in the dataset, we will perturb it to multiple noisy materials by a sequence of Gaussian noises with different noise magnitude levels $\{\sigma_t\}_{t=1}^T$, in which $\sigma_1 > \sigma_2 > ... > \sigma_T$. Let the noisy material at the $t$-th noise level be $\widetilde{\boldsymbol{M}} = \left(\boldsymbol{A}, \widetilde{\boldsymbol{P}}, \boldsymbol{L}\right)$, the noisy coordinate matrix $\widetilde{\boldsymbol{P}} = [\tilde{\boldsymbol{p}}_1, ..., \tilde{\boldsymbol{p}}_n]$ is obtained by adding Gaussian noise $\mathcal{N}(\boldsymbol{0}, \sigma_t^2 \boldsymbol{I})$ to $\boldsymbol{P}$. The score model $\boldsymbol{s}_\theta(\cdot)$ is trained to predict the denoising score matrix $\nabla_{\widetilde{\boldsymbol{P}}} \log p_{\sigma_t}\left(\widetilde{\boldsymbol{P}}|\boldsymbol{P}, \boldsymbol{A}, \boldsymbol{L}\right) = [\tilde{\boldsymbol{s}}_1, ..., \tilde{\boldsymbol{s}}_n]$ from $\widetilde{\boldsymbol{M}}$. Here, each $\tilde{\boldsymbol{s}}_i$ is calculated in a similar way as in Equation (3) with the approximated distance score $\tilde{s}_{i,j,\boldsymbol{k}} = \nabla_{\tilde{d}_{i,j,\boldsymbol{k}}} \log p_{\sigma_t}\left(\widetilde{\boldsymbol{P}}|\boldsymbol{P}, \boldsymbol{A}, \boldsymbol{L}\right)$, where $\tilde{d}_{i,j,\boldsymbol{k}} = ||\tilde{\boldsymbol{p}}_i + \boldsymbol{L}\boldsymbol{k} - \tilde{\boldsymbol{p}}_j||_2$. Motivated from

**Algorithm 1:** Langevin Dynamics Sampling Algorithm for Coordinate Generation

---

**Input:** $A$, $L$, atom number $n$, noise magnitudes $\{\sigma_t\}_{t=1}^{T}$, step size $\epsilon$, step number $q$, score model $s_\theta(\cdot)$
Sample $F_0 \in \mathbb{R}^{3 \times n}$ from uniform distribution $U(0, 1)$ and set $P_0 = LF_0$
**for** $t = 1$ **to** $T$ **do**
    $P_t = P_{t-1}$
    $\alpha_t = \epsilon \cdot \sigma_t^2 / \sigma_T^2$
    **for** $j = 1$ **to** $q$ **do**
        $M_t = (A, P_t, L)$
        Sample $Z \in \mathbb{R}^{3 \times n}$ from $\mathcal{N}(\mathbf{0}, I)$
        Obtain the approximated edge distance score from $s_\theta(\cdot)$ which takes $M_{t-1}$ as inputs
        Obtain $s_\theta(M_{t-1}) = [s_1, ..., s_n]$ where each $s_i$ is computed by Equation (3) with edge distance scores.
        $P_t = P_t + \alpha_t s_\theta(M_t) + \sqrt{2\alpha_t} Z$
    **end for**
**end for**
Output $P_T$ as the finally generated coordinate matrix

---

the strategies of existing methods [30, 42, 55], we model the distribution of $\tilde{d}_{i,j,\boldsymbol{k}}$ as a Gaussian distribution $\mathcal{N}\left(\hat{d}_{i,j,\boldsymbol{k}}, \hat{\sigma}_t^2\right)$, where the computation process of the mean $\hat{d}_{i,j,\boldsymbol{k}}$ and the variance $\hat{\sigma}_t^2$ is described in Appendix B. We can calculate $\tilde{s}_{i,j,\boldsymbol{k}}$ directly by taking the derivative of the mathematical formula of this Gaussian distribution. The score model $s_\theta(\cdot)$ is trained to minimize the difference between the output $o_{i,j,\boldsymbol{k}}$ from $s_\theta(\cdot)$ and $\tilde{s}_{i,j,\boldsymbol{k}}$ for every edge $(i, j, \boldsymbol{k})$ in $\widetilde{M}$ with the loss $\mathcal{L}$:

$$\mathcal{L} = \frac{1}{2T} \sum_{t=1}^{T} \hat{\sigma}_t^2 \mathbb{E}_{\widetilde{M}, M} \left[ \sum_{(i,j,\boldsymbol{k}) \in \boldsymbol{E}\left(\widetilde{M}\right)} \left\| \frac{o_{i,j,\boldsymbol{k}}}{\hat{\sigma}_t} + \frac{\tilde{d}_{i,j,\boldsymbol{k}} - \hat{d}_{i,j,\boldsymbol{k}}}{\hat{\sigma}_t^2} \right\|_2^2 \right].$$

After the score model is trained, we can use it to generate coordinate matrix $P$ from given $A$ and $L$ with annealed Langevin dynamics sampling algorithm [49]. Specifically, we first randomly initializes a coordinate matrix $P_0$, then starting from $M_0 = (A, P_0, L)$, we iteratively update $M_{t-1} = (A, P_{t-1}, L)$ to $M_t = (A, P_t, L)$ ($t = 1, ..., T$) with the approximated score matrix $s_\theta(M_{t-1})$ by running multiple Langevin dynamics sampling steps. See Algorithm 1 for the detailed coordinate generation algorithm pseudocodes.

### 3.5 Discussions and Relations with Prior Methods

**Advantages and limitations.** In our method, the atom type vector and lattice matrix are first transformed to items that are invariant to symmetry transformations, then a VAE model is used to capture the distributions of these items. In addition, our method adopts powerful score-based diffusion models to generate atom coordinates, and a novel probabilistic modeling framework for score approximation is employed to satisfy all symmetry-related restrictions. With all these strategies, our method successfully incorporates physical symmetries of periodic materials into generative models, thereby capturing the underlying distributions of material structures more effectively. We will show these advantages through experiments in Section 4. The major limitations of our method lie in that the speed of generating atom coordinates with score-based diffusion models is slow because running thousands of Langevin dynamics sampling steps is needed, and our method cannot be applied to non-periodic materials. We leave more discussions about limitations and broader impacts in Appendix C.

**Motivations in Model Design.** The model design of our SyMat method has the following three motivations. (1) First, the structures of atom types and lattices are relatively simple and their symmetry-invariant representations (*i.e.*, atom type sets, lattice lengths and angles) are low-dimensional vectors. Their distributions are simple enough to be captured by VAE models. Also, VAE models are faster than diffusion models in generation speed. Hence, we use VAE models for atom type and lattice generation. (2) Second, the structure of atom coordinates is much more complicated than atom types or lattices, so we use more powerful score-based diffusion models to generate them. More

importantly, using score-based diffusion models for atom coordinates enables us to trickily convert the prediction of coordinate scores to that of interatomic distance scores by the chain rule of derivatives. This strategy maintains invariance to rotation, translation and periodic transformations. We think it is hard to develop such a symmetry-aware probabilistic modeling for other generative models. Hence, we use score-based diffusion models for coordinate generation. (3) Third, SphereNet is chosen as the backbone network of our model. SphereNet is a 3D graph neural network for 3D graph representation learning. It has powerful capacities in 3D structure feature extraction because it considers complete 3D geometric information contained in pairwise distances, line angles and plane angles (torsion angles). Also, the features extracted by SphereNet are invariant to rotation, translation and periodic transformations when input 3D graphs are created by multi-graph method. Because of its powerful capacity and symmetry-aware feature extraction, we use SphereNet as the backbone network.

**Relations with prior methods.** Here we will discuss several studies that are related to our work. Some studies [42, 30] have also applied score-based diffusion models to generate atom coordinates from molecular graphs, but there are not periodic structures in 3D molecules. Differently, keeping invariant to periodic transformations is significant in periodic material generation, and our score-based diffusion model uses periodic invariant multi-graph representations of materials as inputs to satisfy this property. Besides, compared with a recently proposed periodic material generation method CDVAE [52], our SyMat has two major differences. (1) First, to generate the atom type vector, CDVAE generates total atom numbers and compositions (the proportion of atom numbers for every atom type to the total atom number) by a VAE model, in which compositions are vectors composed of continuous numbers. While compositions have complicated structures and are hard to predict, SyMat uses relatively simpler atom type sets as generation targets and the prediction of them can be easily converted to integer prediction or classification problems (see Section 3.3). (2) Second, in coordinate generation, CDVAE also uses the denoising score matching framework, but the denoising score matrix $\nabla_{\widetilde{P}} \log p_{\sigma_t}(\widetilde{P}|P, A, L)$ is calculated as the direct difference between $\widetilde{P}$ and a coordinate matrix obtained by aligning $P$, which makes $\nabla_{\widetilde{P}} \log p_{\sigma_t}(\widetilde{P}|P, A, L)$ not invariant to translations on $\widetilde{P}$. However, SyMat calculates it from the edge distance score functions to ensure invariance to all symmetry transformations (see Section 3.4). In addition, another two studies, PGD-VAE [47] and DiffCSP [18], are also related to our SyMat approach. PGD-VAE proposes a VAE model for periodic graph generation. However, PGD-VAE is fundamentally different from SyMat in that it only generates 2D periodic topology graphs without 3D structures, while SyMat is developed for 3D periodic material structures. PGD-VAE cannot be applied to 3D periodic materials so we do not compare SyMat with PGD-VAE in our experiments. DiffCSP proposes a novel method for generating 3D periodic material structures from their atom types. Different from SyMat, DiffCSP uses denoising diffusion models to jointly generate lattices and atom coordinates. In addition, DiffCSP does not generate atom types of periodic materials, but lattices and atom coordinates from the input atom types, so DiffCSP cannot be applied to design novel periodic materials from scratch. Hence, we do not compare SyMat with DiffCSP in our experiments as all our experiments evaluate the performance of different methods in generating novel periodic materials.

# 4 Experiments

In this section, we evaluate our proposed SyMat method in two periodic material generation tasks, including random generation and property optimization. We show that our proposed SyMat can achieve promising performance in both tasks.

## 4.1 Experimental Setup

**Data.** We evaluate SyMat on three benchmark datasets curated by Xie et al. [52], including Perov-5 [3, 2], Carbon-24 [35], and MP-20 [17]. Perov-5 is a dataset with a collection of 18,928 perovskite materials. The chemical element compositions of all materials in Perov-5 can be denoted by a general chemical formula $ABX_3$, which means that there are 3 different atom types and 5 atoms in each unit cell. The Carbon-24 dataset we used has 10,153 materials. All these material structures consist of only carbon elements and have 6 - 24 atoms in each unit cell, and the 3D structures of them are obtained by DFT simulation. MP-20 is a dataset curated from Materials Project [17] library. It includes 45,231 materials with various structures and compositions, and in all materials, there exist at

Table 1: Random generation performance on Perov-5, Carbon-24, and MP-20 datasets. Here ↑ means higher metric values lead to better performance, while ↓ means the opposite. Because all materials in the Carbon-24 dataset are composed of only carbon atoms, all methods easily achieve 100% in composition validity and 0.0 in # element EMD so we do not use these two metrics. Also, Cond-DFC-VAE can only be used to Perov-5 dataset in which materials have cubic structures. **Bold** and underline numbers highlight the best and second best performance, respectively.

| Dataset | Method | Composition validity↑ | Structure validity↑ | # Element EMD↓ | Density EMD↓ | Energy EMD↓ | COV-R↑ | COV-P↑ |
|---|---|---|---|---|---|---|---|---|
| Perov-5 | FTCP | 54.24% | 0.24% | 0.6297 | 10.27 | 156.0 | 0.00% | 0.00% |
| | Cond-DFC-VAE | 82.95% | 73.60% | 0.8373 | 2.268 | 4.111 | 77.80% | 12.38% |
| | G-SchNet | 98.79% | 99.92% | 0.0368 | 1.625 | 4.746 | 0.25% | 0.37% |
| | P-G-SchNet | **99.13%** | 79.63% | 0.4552 | 0.2755 | 1.388 | 0.56% | 0.41% |
| | CDVAE | 98.59% | **100.00%** | 0.0628 | **0.1258** | **0.0264** | 99.50% | **98.93%** |
| | SyMat (ours) | 97.40 % | **100.00%** | **0.0177** | 0.1893 | 0.2364 | **99.68%** | 98.64% |
| Carbon-24 | FTCP | – | 0.08% | – | 5.206 | 19.05 | 0.00% | 0.00% |
| | G-SchNet | – | 99.94% | – | 0.9427 | 1.320 | 0.00% | 0.00% |
| | P-G-SchNet | – | 48.39% | – | 1.533 | 134.7 | 0.00% | 0.00% |
| | CDVAE | – | **100.00%** | – | 0.1407 | **0.2850** | **100.00%** | **99.98%** |
| | SyMat (ours) | – | **100.00%** | – | **0.1195** | 3.9576 | **100.00%** | 97.59% |
| MP-20 | FTCP | 48.37% | 1.55% | 0.7363 | 23.71 | 160.9 | 5.26% | 0.23% |
| | G-SchNet | 75.96% | 99.65% | 0.6411 | 3.034 | 42.09 | 41.68% | 99.65% |
| | P-G-SchNet | 76.40% | 77.51% | 0.6234 | 4.04 | 2.448 | 44.89% | 99.76% |
| | CDVAE | 86.70% | **100.00%** | 1.432 | 0.6875 | **0.2778** | **99.17%** | 99.64% |
| | SyMat (ours) | **88.26%** | **100.00%** | **0.5067** | **0.3805** | 0.3506 | 98.97% | **99.97%** |

most 20 atoms in each unit cell. For all three datasets, we split them with a ratio of 3:1:1 as training, validation, and test sets in our experiments.

**Tasks.** We evaluate the performance of SyMat in random generation and property optimization tasks. In the random generation task, SyMat generates novel periodic materials from randomly sampled latent variables and we employ a variety of validity and statistic metrics to evaluate the quality of the generated periodic materials. In the property optimization task, SyMat is evaluated by the ability of discovering novel periodic materials with good properties. Besides, to show that latent representations are informative enough to reconstruct periodic materials, we also evaluate the performance by the material reconstruction task. See Appendix D.2 for experiment results of this task.

**Baselines.** We compare SyMat with two early periodic material generation methods FTCP [36] and Cond-DFC-VAE [6], which generate material structures through generating Fourier-transformed crystal property matrices and 3D voxel images with VAE models, respectively. Note that Cond-DFC-VAE can only be used to generate cubic systems, so it is only used for Perov-5 dataset in which materials have cubic structures. In addition, we compare with an autoregressive 3D molecule generation method G-SchNet [10], and its variant P-G-SchNet that incorporates periodicity in G-SchNet. The latest periodic material generation method CDVAE [52] is also compared with our method. Note that all these baseline methods are compared with SyMat in the random generation task, but G-SchNet and P-G-SchNet are not compared in the property optimization task because they cannot produce latent representations with fixed dimensions for different periodic materials.

## 4.2 Random Generation

**Metrics.** In the random generation task, we adopt seven metrics to evaluate physical and statistic properties of materials generated by our method and baseline methods. We evaluate the composition validity and structure validity, which are the percentages of generated materials with valid atom type vectors and 3D structures, respectively. Here, an atom type vector is considered as valid if its overall charge computed by SMACT package [8] is neutral, and following Court et al. [6], a 3D structure is considered as valid if the distances between any pairwise atoms is larger than 0.5Å. In addition, we evaluate the similarity between the generated materials and the materials in the test set by five statistic metrics, including the earth mover's distance (EMD) between the distributions in chemical element number, density (g/cm$^3$), and energy predicted by a GNN model. Also, we evaluate the percentage of the test set materials that cover at least one of the generated materials (COV-R), and the percentage of generated materials that cover at least one of the test set materials (COV-P). All seven metrics are evaluated on 10,000 generated periodic materials. See Appendix D.1 for more information about evaluation metrics and experimental settings in the random generation task.

Table 2: Property optimization performance on Perov-5, Carbon-24, and MP-20 datasets. We report the success rate (SR), *i.e.*, the rate of periodic materials that can achieve top 5%, 10%, and 15% of the property distribution in the dataset after being optimized. Cond-DFC-VAE can only be used to Perov-5 dataset in which materials have cubic structures. **Bold** and underline numbers highlight the best and second best performance, respectively.

| Method | Perov-5 | | | Carbon-24 | | | MP-20 | | |
|---|---|---|---|---|---|---|---|---|---|
| | SR5 | SR10 | SR15 | SR5 | SR10 | SR15 | SR5 | SR10 | SR15 |
| FTCP | 0.06 | 0.11 | 0.16 | 0.00 | 0.00 | 0.00 | 0.02 | 0.04 | 0.05 |
| Cond-DFC-VAE | 0.55 | 0.64 | 0.69 | – | – | – | – | – | – |
| CDVAE | 0.52 | 0.65 | 0.79 | 0.00 | 0.06 | 0.06 | 0.78 | 0.86 | 0.90 |
| SyMat (ours) | **0.73** | **0.80** | **0.87** | **0.06** | **0.13** | **0.13** | **0.92** | **0.97** | **0.97** |

**Results.** The random generation performance results of all methods on three datasets are presented in Table 1. Among the total 19 metrics over three datasets in the table, our SyMat method achieves top-1 performance in 11 metrics and top-2 performance in 17 metrics. Particularly, SyMat can achieve excellent performance in composition and structure validity, which demonstrates that SyMat can accurately learn the rules of forming chemically valid and stable periodic materials so it generates chemically valid periodic materials with high chances. Overall, SyMat achieves promising performance in the random generation task, showing that SyMat is an effective approach to capture the distribution of complicated periodic material structures.

We visualize some generated periodic materials in Figure 3 of Appendix E, and use additional metrics to evaluate the diversity of the generated periodic materials in Appendix D.3. To demonstrate the effectiveness of coordinate generation module in SyMat, we conduct an ablation study using several evaluation metrics in the random generation task and present this part in Appendix D.4.

## 4.3 Property Optimization

**Metrics.** In the property optimization task, we evaluate the ability of SyMat in discovering novel periodic materials with low energies. We jointly train the SyMat model with an energy prediction model that predicts the energies from the latent representations. Afterwards, we follow the evaluation procedure in [52] to optimize the latent representations of testing materials with 5000 gradient descent steps, decode 100 periodic materials from the optimized latent representations to periodic materials, and report the success rates (SR), *i.e.*, the rate of decoded materials whose properties are in the top 5%, 10%, and 15% of the energy distribution in the dataset. We use the same evaluation procedure to baseline methods and compare them with SyMat in optimization success rates.

**Results.** The property optimization performance results of all methods on three datasets are summarized in Table 2. On three datasets, SyMat outperforms all baseline methods in optimization success rates, showing that SyMat has the highest chances to generate periodic materials with low energies. This demonstrates that SyMat models have strong capacities in discovering novel periodic materials with good properties. The good performance of SyMat in the property optimization method shows that it can be an effective tool for designing novel periodic materials with desired properties.

We visualize the changes of material structures and energies of some periodic materials after property optimization in Figure 4 of Appendix E. We also visualize the material energy distribution before and after property optimization by SyMat models in Figure 5 of Appendix E.

## 5 Conclusion

We propose SyMat, a novel deep generative model for periodic material generation. SyMat achieves invariance to physical symmetry transformations of periodic materials, including permutation, rotation, translation, and periodic transformations. In SyMat, atom types and lattices of materials are generated in the form of atom type sets, lattice lengths and lattice angles with a VAE model. Additionally, SyMat uses a score-based diffusion model based on a symmetry-aware coordinate diffusion process to generate atom coordinates. Experiments on random generation and property optimization tasks show that SyMat is a promising approach for discovering novel materials. In the future, we will study the problem of accelerating atom coordinate generation in SyMat.

## Acknowledgments and Disclosure of Funding

We thank Xiang Fu for his help on providing technical details about running baseline methods. This work was supported in part by National Science Foundation under grants IIS-1908220 and IIS-2006861.

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

# A Proof of Propositions

## A.1 Proof of Proposition 3.1

We first prove Proposition 3.1, *i.e.*, if the material distribution $p(\boldsymbol{M})$ is invariant to permutation, rotation, translation, and periodic transformations described in Section 3.1, the score matrix $\nabla_{\boldsymbol{P}} \log p(\boldsymbol{P}|\boldsymbol{A}, \boldsymbol{L})$ will be invariant to translation and periodic transformations, and equivariant to permutation and rotation transformations.

(1) For $\boldsymbol{M}' = (\boldsymbol{A}', \boldsymbol{P}', \boldsymbol{L})$ where $\boldsymbol{A}'$ and $\boldsymbol{P}'$ are obtained by permuting the elements of $\boldsymbol{A}$ and column vectors of $\boldsymbol{P}$ with the same order, from $p(\boldsymbol{M}) = p(\boldsymbol{M}')$, we know that $p(\boldsymbol{P}'|\boldsymbol{A}', \boldsymbol{L}) = p(\boldsymbol{P}|\boldsymbol{A}, \boldsymbol{L})$ holds, so $\nabla_{\boldsymbol{P}'} \log p(\boldsymbol{P}'|\boldsymbol{A}', \boldsymbol{L})$ actually equals to the matrix obtained by permuting the column vectors of $\nabla_{\boldsymbol{P}} \log p(\boldsymbol{P}|\boldsymbol{A}, \boldsymbol{L})$ with the same permutation order. In other words, $\nabla_{\boldsymbol{P}} \log p(\boldsymbol{P}|\boldsymbol{A}, \boldsymbol{L})$ is permutation-equivariant.

(2) For $\boldsymbol{M}' = (\boldsymbol{A}, \boldsymbol{P}', \boldsymbol{L}')$ obtained by rotating $\boldsymbol{M} = (\boldsymbol{A}, \boldsymbol{P}, \boldsymbol{L})$ with the rotation matrix $\boldsymbol{Q}$ and translating it with the translation vector $\boldsymbol{b}$, because $p(\boldsymbol{M}') = p(\boldsymbol{M})$ and $\boldsymbol{P}' = \boldsymbol{Q}\boldsymbol{P} + \boldsymbol{b}\boldsymbol{1}^T$, we have $p(\boldsymbol{P}'|\boldsymbol{A}, \boldsymbol{L}') = p(\boldsymbol{P}|\boldsymbol{A}, \boldsymbol{L})$ and $\nabla_{\boldsymbol{P}'} \log p(\boldsymbol{P}'|\boldsymbol{A}, \boldsymbol{L}') = \boldsymbol{Q}\nabla_{\boldsymbol{P}} \log p(\boldsymbol{P}|\boldsymbol{A}, \boldsymbol{L})$. In other words, $\nabla_{\boldsymbol{P}} \log p(\boldsymbol{P}|\boldsymbol{A}, \boldsymbol{L})$ is translation-invariant and rotation-equivariant.

(3) For $\boldsymbol{M}^K = (\boldsymbol{A}, \boldsymbol{P}^K, \boldsymbol{L})$ where $\boldsymbol{P}^K = \boldsymbol{P} + \boldsymbol{L}\boldsymbol{K}$ is obtained by periodic transformation, since $p(\boldsymbol{M}) = p(\boldsymbol{M}^K)$, we can easily find that $p(\boldsymbol{P}^K|\boldsymbol{A}, \boldsymbol{L}) = p(\boldsymbol{P}|\boldsymbol{A}, \boldsymbol{L})$ and $\nabla_{\boldsymbol{P}^K} \log p(\boldsymbol{P}^K|\boldsymbol{A}, \boldsymbol{L}) = \nabla_{\boldsymbol{P}} \log p(\boldsymbol{P}|\boldsymbol{A}, \boldsymbol{L})$ hold. In other words, $\nabla_{\boldsymbol{P}} \log p(\boldsymbol{P}|\boldsymbol{A}, \boldsymbol{L})$ is invariant to periodic transformations.

## A.2 Proof of Proposition 3.2

We next prove Proposition 3.2, *i.e.*, if $s_{i,j,\boldsymbol{k}}$ is always invariant to permutation, rotation, translation, and periodic transformations described in Section 3.1, the score matrix $\nabla_{\boldsymbol{P}} \log p(\boldsymbol{P}|\boldsymbol{A}, \boldsymbol{L})$ computed by Equation (3) will always be invariant to translation and periodic transformations, and equivariant to permutation and rotation transformations.

(1) The score matrix $\nabla_{\boldsymbol{P}} \log p(\boldsymbol{P}|\boldsymbol{A}, \boldsymbol{L}) = [\boldsymbol{s}_1, ..., \boldsymbol{s}_n]$ is formed by stacking coordinate score vector $\boldsymbol{s}_i$ according to the atom order, and this stacking operation is clearly equivariant to permutations. Also, from Equation (3), we can easily find that if $s_{i,j,\boldsymbol{k}}$ is always invariant to permutation, the computed coordinate score vector $\boldsymbol{s}_i$ for each coordinate $\boldsymbol{p}_i$ will also be invariant to permutations. Hence, $\nabla_{\boldsymbol{P}} \log p(\boldsymbol{P}|\boldsymbol{A}, \boldsymbol{L})$ from $\boldsymbol{s}_1, ..., \boldsymbol{s}_n$ is equivariant to permutations.

(2) For any $\boldsymbol{Q} \in \mathbb{R}^{3 \times 3}, \boldsymbol{Q}^T\boldsymbol{Q} = \boldsymbol{I}$ and $\boldsymbol{b} \in \mathbb{R}^3$, let $\boldsymbol{P}' = [\boldsymbol{p}'_1, ..., \boldsymbol{p}'_n] = \boldsymbol{Q}\boldsymbol{P} + \boldsymbol{b}\boldsymbol{1}^T$ where $\boldsymbol{p}'_i = \boldsymbol{Q}\boldsymbol{p}_i + \boldsymbol{b}, i = 1, ..., n$, $\boldsymbol{L}' = \boldsymbol{Q}\boldsymbol{L}$, and $\boldsymbol{M}' = (\boldsymbol{A}, \boldsymbol{P}', \boldsymbol{L}')$. We can calculate the score vector $\boldsymbol{s}'_i$ for $\boldsymbol{p}'_i$ in the way similar to Equation (3) as

$$
\begin{aligned}
\boldsymbol{s}'_i &= \nabla_{\boldsymbol{p}'_i} \log p(\boldsymbol{P}'|\boldsymbol{A}, \boldsymbol{L}') \\
&= \sum_{(j,\boldsymbol{k}) \in \mathcal{N}'(i)} \nabla_{d'_{i,j,\boldsymbol{k}}} \log p(\boldsymbol{P}'|\boldsymbol{A}, \boldsymbol{L}') \cdot \nabla_{\boldsymbol{p}'_i} d'_{i,j,\boldsymbol{k}} \\
&= \sum_{(j,\boldsymbol{k}) \in \mathcal{N}'(i)} s'_{i,j,\boldsymbol{k}} \cdot \frac{\boldsymbol{p}'_i + \boldsymbol{L}'\boldsymbol{k} - \boldsymbol{p}'_j}{||\boldsymbol{p}'_i + \boldsymbol{L}'\boldsymbol{k} - \boldsymbol{p}'_j||_2}, \\
&= \sum_{(j,\boldsymbol{k}) \in \mathcal{N}'(i)} s'_{i,j,\boldsymbol{k}} \cdot \frac{\boldsymbol{Q}\boldsymbol{p}_i + \boldsymbol{b} + \boldsymbol{Q}\boldsymbol{L}\boldsymbol{k} - \boldsymbol{Q}\boldsymbol{p}_j - \boldsymbol{b}}{||\boldsymbol{Q}\boldsymbol{p}_i + \boldsymbol{b} + \boldsymbol{Q}\boldsymbol{L}\boldsymbol{k} - \boldsymbol{Q}\boldsymbol{p}_j - \boldsymbol{b}||_2}, \\
&= \sum_{(j,\boldsymbol{k}) \in \mathcal{N}(i)} s_{i,j,\boldsymbol{k}} \cdot \frac{\boldsymbol{Q}(\boldsymbol{p}_i + \boldsymbol{L}\boldsymbol{k} - \boldsymbol{p}_j)}{||\boldsymbol{p}_i + \boldsymbol{L}\boldsymbol{k} - \boldsymbol{p}_j||_2}, \\
&= \boldsymbol{Q}\boldsymbol{s}_i,
\end{aligned}
$$

where $\mathcal{N}'(i) = \{(j, \boldsymbol{k}) : (i, j, \boldsymbol{k}) \in \boldsymbol{E}(\boldsymbol{M}')\}, \mathcal{N}(i) = \{(j, \boldsymbol{k}) : (i, j, \boldsymbol{k}) \in \boldsymbol{E}(\boldsymbol{M})\}$. Note that the second to the last equation holds because $||\boldsymbol{Q}\boldsymbol{x}||_2 = ||\boldsymbol{x}||_2$ holds for any $\boldsymbol{x} \in \mathbb{R}^3$ if $\boldsymbol{Q}^T\boldsymbol{Q} = \boldsymbol{I}$, and the edge set of multi-graph representation and $s_{i,j,\boldsymbol{k}}$ is invariant to any rotation and translation transformations, hence $\boldsymbol{E}(\boldsymbol{M}') = \boldsymbol{E}(\boldsymbol{M})$ and $s'_{i,j,\boldsymbol{k}} = s_{i,j,\boldsymbol{k}}$. So far, we have shown that $\nabla_{\boldsymbol{P}} \log p(\boldsymbol{P}|\boldsymbol{A}, \boldsymbol{L})$ is invariant to translations and equivariant to rotations.

(3) For any $\boldsymbol{K} \in \mathbb{Z}^{3\times3}$, let $\boldsymbol{P}^{\boldsymbol{K}} = [\boldsymbol{p}_1', ..., \boldsymbol{p}_n']\boldsymbol{P}+\boldsymbol{L}\boldsymbol{K}$ and $\boldsymbol{M}^{\boldsymbol{K}} = \left(\boldsymbol{A}, \boldsymbol{P}^{\boldsymbol{K}}, \boldsymbol{L}\right)$. Because the multi-graph method is strictly invariant to periodic transformations, there must exist a bijective mapping $f : \boldsymbol{E}(\boldsymbol{M}) \to \boldsymbol{E}\left(\boldsymbol{M}^{\boldsymbol{K}}\right)$ such that for any $(i, j, \boldsymbol{k}) \in \boldsymbol{E}(\boldsymbol{M})$, $(i, j, \boldsymbol{k}') = f(i, j, \boldsymbol{k})$ satisfies that $\boldsymbol{p}_i + \boldsymbol{L}\boldsymbol{k} - \boldsymbol{p}_j = \boldsymbol{p}_i' + \boldsymbol{L}\boldsymbol{k}' - \boldsymbol{p}_j'$. Also, because $s_{i,j,\boldsymbol{k}}$ is invariant to periodic transformations, we can easily find that calculating the score vectors for $\boldsymbol{p}_i$ and $\boldsymbol{p}_i'$ by Equation (3) are actually summing up the same set of vectors, so these two score vectors equal and then $\nabla_{\boldsymbol{P}} \log p(\boldsymbol{P}|\boldsymbol{A}, \boldsymbol{L})$ is invariant to periodic transformations.

# B    Additional Implementation Details

**SphereNet Model.** In SyMat, we use SphereNet model [29] as the backbone GNN architecture for both the encoder model in the VAE for atom type and lattice generation and the score model in the score-based diffusion model for atom coordinate generation. SphereNet is a powerful 3D GNN model proposed for 3D molecular property prediction, but in our method, we will use the multi-graph representations of materials as inputs to SphereNet models. Generally, SphereNet model first initializes the embedding vectors of nodes and edges with a look-up atom type embedding vectors and spherical basis functions, then employ multiple message passing [11] layers to iteratively update the node and edge embeddings. Note that this overall process of initializing and updating embeddings is always invariant to all symmetry transformations described in Section 3.1. The encoder model of VAE model will use a sum pooling over the final node embeddings to obtain a global representation vector, which is passed into an MLP to obtain $z^A$ and $z^L$. The score model will use another MLP to predict the edge distance score scalar from the final edge embeddings. Note that we use two independent SphereNet models whose parameters are not shared for the VAE encoder model and score model. We use the implementations of SphereNet in DIG package [26].

**Obtaining lattice matrix from lattice lengths and angles.** From the lattice lengths $\boldsymbol{\ell} = [\ell_1, \ell_2, \ell_3]$ and lattice angles $\boldsymbol{\phi} = [\phi_{12}, \phi_{23}, \phi_{13}]$, let $\gamma = \arccos \frac{\cos \phi_{23} \cos \phi_{13} - \cos \phi_{12}}{\sin \phi_{23} \sin \phi_{13}}$, one lattice matrix $\boldsymbol{L} = [\boldsymbol{\ell}_1, \boldsymbol{\ell}_2, \boldsymbol{\ell}_3]$ can be computed as

$$\boldsymbol{\ell}_1 = [\ell_1 \sin \phi_{13}, 0, \ell_1 \cos \phi_{13}]^T,$$
$$\boldsymbol{\ell}_2 = [-\ell_2 \sin \phi_{23} \cos \gamma, \ell_2 \sin \phi_{23} \sin \gamma, \ell_2 \cos \phi_{23}]^T,$$
$$\boldsymbol{\ell}_3 = [0, 0, \ell_3]^T.$$

**Calculating the denoising score matrix.** In the training of score-based diffusion model in SyMat, we add Gaussian noise with noise magnitude $\sigma_t$ to the coordinate matrix of a material $\boldsymbol{M} = (\boldsymbol{A}, \boldsymbol{P}, \boldsymbol{L})$ in the dataset to obtain a noisy material $\widetilde{\boldsymbol{M}} = \left(\boldsymbol{A}, \widetilde{\boldsymbol{P}}, \boldsymbol{L}\right)$, and the score model $\boldsymbol{s}_\theta(\cdot)$ is optimized to predict the denoising score matrix $\nabla_{\widetilde{\boldsymbol{P}}} \log p_{\sigma_t} \left(\widetilde{\boldsymbol{P}} | \boldsymbol{P}, \boldsymbol{A}, \boldsymbol{L}\right)$. Then the score model $\boldsymbol{s}_\theta(\cdot)$ tries to predict the edge distance score $\tilde{s}_{i,j,\boldsymbol{k}} = \nabla_{\tilde{d}_{i,j,\boldsymbol{k}}} \log p_{\sigma_t} \left(\widetilde{\boldsymbol{P}} | \boldsymbol{P}, \boldsymbol{A}, \boldsymbol{L}\right)$ for every edge $(i, j, \boldsymbol{k})$ in the multi-graph representation of $\widetilde{\boldsymbol{M}}$. Motivated from existing methods [30, 42, 55] in molecule generation, we can assume that the distribution of edge distance $\tilde{d}_{i,j,\boldsymbol{k}}$ in $\widetilde{\boldsymbol{M}}$ is a Gaussian distribution centered around the edge distance of the same edge in $\boldsymbol{M}$. Practically, we find that using a new coordinate matrix $\widehat{\boldsymbol{P}} = [\hat{\boldsymbol{p}}_1, ..., \hat{\boldsymbol{p}}_n]$ obtained by aligning $\boldsymbol{P} = [\boldsymbol{p}_1, ..., \boldsymbol{p}_n]$ w.r.t. $\widetilde{\boldsymbol{P}} = [\tilde{\boldsymbol{p}}_1, ..., \tilde{\boldsymbol{p}}_n]$ to calculate the corresponding edge distance in $\boldsymbol{M}$ can achieve the best performance. Specifically, we follow alignment procedure as in CDVAE [52] and calculate each $\hat{\boldsymbol{p}}_i$ as $\hat{\boldsymbol{p}}_i = \boldsymbol{p}_i + \boldsymbol{L}\boldsymbol{u}$, where $\boldsymbol{u} = \arg\min_{\boldsymbol{v} \in \mathbb{Z}^3} ||\boldsymbol{p}_i + \boldsymbol{L}\boldsymbol{v} - \tilde{\boldsymbol{p}}_i||_2$. The mean $\hat{d}_{i,j,\boldsymbol{k}}$ of the Gaussian distribution of the edge distance $\tilde{d}_{i,j,\boldsymbol{k}}$ is calculated as $\hat{d}_{i,j,\boldsymbol{k}} = ||\hat{\boldsymbol{p}}_i + \boldsymbol{L}\boldsymbol{k} - \hat{\boldsymbol{p}}_j||_2$. Besides, we assume the Gaussian distributions of $\hat{d}_{i,j,\boldsymbol{k}}$ for every edge have the same standard deviations $\sigma_t$, which is pre-computed by manually perturbing all materials in the dataset to noisy materials, collecting the edge distances in all noisy materials, and calculating the empirical standard deviation as $\sigma_t$.

# C  More Discussions

**Limitations.** Generally, there are three major limitations in our proposed SyMat approach. (1) First, SyMat may take as long as an hour to generate atom coordinates of periodic materials with score-based diffusion models, since this process requires to run thousands of Langevin dynamics sampling steps. Actually, this is a common limitation for many other molecule or material generation methods [42, 30, 52] using score-based diffusion models. Nonetheless, we argue that compared with the huge time cost of laborious lab experiments or DFT based calculation, the time cost of SyMat is relatively lower and acceptable. We will explore accelerating SyMat with SDE based diffusion models [45, 23] in the future. (2) Second, SyMat is designed to maintain invariance to periodic transformations, so it cannot be applied to non-periodic materials as this will lead to incorrect probabilistic modeling. Though some materials are non-periodic, such as amorphous solids and polymers, we focus on periodic materials because they occupy a large part of materials used in real-world applications. We believe that developing non-periodic material generation method need to rely on effective periodic material generation methods, but the problem of generating relatively simpler periodic materials still remains challenging and largely unsolved. Hence, we will focus on generating periodic materials in this work and leaves the problem of generating non-periodic materials to the future. (3) Third, our work follows previous material generation studies [52] to evaluate the quality of the generated materials by some basic metrics, such as composition and structure validity. However, they do not use metrics to evaluate the synthesizability of the generated materials, though these metrics are useful for practical applications. In the future, we will collaborate with experts in material science to come up with metrics for evaluating synthesizability of the materials generated by our method.

**Broader impacts.** In this work, we propose a novel method for periodic material generation. Our work can be used to design novel materials with desired properties for many real-world applications, such as batteries or solar cells. Nonetheless, our method may unexpectedly generate materials that produce negative impacts to human life. For example, manufacturing some materials generated by our model may cause environmental pollution. Hence, we believe strict evaluation or validation process need to be done to estimate the related environmental or other social impacts before using the materials generated by our model to real-world scenarios.

# D  Additional Experiment Details and Results

## D.1  Additional Details in Experimental Settings

In our SyMat model, the SphereNet model is composed of 4 message passing layers and the hidden size is set to 128. Also, all MLP models in the VAE decoder is composed of 2 linear layers with a ReLU function between them, and the hidden size is set to 256. During training, we set the learning rate to 0.001, the batch size to 128, and the epoch number to 1,000. The total training needs around 3, 5, and 10 hours on a GTX 2080 GPU for Perov-5, Carbon-24, and MP-20 datasets, separately. We use different weights for different loss terms. Specifically, we set the weights of the reconstruction losses for atom type set size, atom types, atom numbers of each atom type, lattice items as 1.0, 30.0, 1.0, 10.0, respectively, and use a weight of 0.01 for KL-distance loss and a weight of 10.0 for denoising score matching loss. In the generation of atom coordinates with Algorithm 1, we set step size $\epsilon$ as 0.0001 and step number $q$ as 100. In addition, we use a geometrically decreasing series between 10 and 0.01 as $\{\sigma_t\}_{t=1}^{T}$ and set $T = 50$. In the random generation task, for COV-R and COV-P metrics, we consider one material cover another material if the distances of their chemical element composition fingerprints and 3D structure fingerprints are smaller than the thresholds $\delta_c$ and $\delta_s$, respectively. For Perov-5 dataset, we use $\delta_c = 6$, $\delta_s = 0.8$; for Carbon-24 dataset, we use $\delta_c = 4$, $\delta_s = 1.0$; for MP-24 dataset, we use $\delta_c = 12$, $\delta_s = 0.6$.

## D.2  Material Reconstruction

In the material reconstruction task, we reconstruct materials in the test sets of Perov-5, Carbon-24 and MP-20 datasets from the encoded latent representations $z^A$ and $z^L$. We evaluate the reconstruction performance by several metrics. First, we evaluate the percentage of materials whose atom types can be exactly reconstructed (Atom Type Match Rate), and the root mean square error in lattice lengths and angles between the target and reconstructed materials (Lattice RMSE). Second, to show the score-based diffusion model has the capacity to approximately reconstruct 3D structures, we evaluate the root mean square error in interatomic distances between the target and reconstructed materials (Distance RMSE). We present the experimental results of SyMat together with baseline methods in Table 3. Results show that our method performs well in atom types, lattices and interatomic distances reconstruction.

Table 3: Reconstruction performance on Perov-5, Carbon-24, and MP-20 datasets. Here ↑ means higher metric values lead to better performance, while ↓ means the opposite. Note that Cond-DFC-VAE can only be used to Perov-5 dataset in which materials have cubic structures. **Bold** and underline numbers highlight the best and second best performance, respectively.

| Dataset | Method | Atom type match rate↑ | Lattice RMSE↓ | Distance RMSE↓ |
|---|---|---|---|---|
| Perov-5 | FTCP | **99.67%** | 0.0786 | 0.2953 |
| | Cond-DFC-VAE | 58.92% | 0.0765 | 0.2281 |
| | CDVAE | 98.16% | 0.0231 | 0.1072 |
| | SyMat (ours) | 98.30% | **0.0224** | **0.0723** |
| Carbon-24 | FTCP | **100.00%** | 0.1349 | 0.3759 |
| | CDVAE | **100.00%** | **0.0624** | 0.1745 |
| | SyMat (ours) | **100.00%** | 0.0632 | **0.1186** |
| MP-20 | FTCP | 71.46% | 0.1057 | 0.2062 |
| | CDVAE | 68.32% | 0.0532 | 0.0975 |
| | SyMat (ours) | **72.10%** | **0.0510** | **0.0876** |

## D.3  Diversity Evaluation

We evaluate the diversity of the randomly generated atom types by uniqueness percentage, *i.e.*, the percentage of unique atom types among all the generated atom types. We present the uniqueness percentages of SyMat and baseline methods on Perov-5 and MP-20 datasets in Table 4. Note that Cond-DFC-VAE can only be used to Perov-5 dataset in which materials have cubic structures, and

Carbon-24 is not used here because all materials in Carbon-24 are composed of only carbon atoms, so the uniqueness is expected to be low. Results show that our method performs well in generating materials with diverse atom types.

Table 4: Uniqueness percentages on Perov-5 and MP-20 datasets.

| Dataset | FTCP | Cond-DFC-VAE | G-SchNet | P-G-SchNet | CDVAE | SyMat |
|---------|------|--------------|----------|------------|-------|-------|
| Perov-5 | 72.33% | 80.36% | 98.74% | 98.57% | 98.61% | **99.43%** |
| MP-20 | 79.04% | – | 99.23% | 99.03% | 99.84% | **99.98%** |

## D.4 Ablation Study

We conduct an ablation study experiment to show that the use of distance score matching in our coordinate generation module is significant. We implement a variant of SyMat in which the score model in coordinate generation module is trained by applying the denoising score matching loss directly to coordinates, instead of using our proposed distance score matching loss. We evaluate its performance in the random generation task by the structural validity, COV-R, and COV-P metrics as these metrics evaluate the quality of the generated atom coordinates. We summarize the results in Table 5. From the results, we can clearly find that using coordinate score matching always achieves lower COV-R and COV-P than using distance score matching. This demonstrates that distance score matching can more accurately capture the distribution of material structures, so the generated periodic materials are more similar to periodic materials in datasets, thereby achieving higher COV-R and COV-P.

Table 5: Comparison of SyMat with coordinate score matching and distance score matching. Here $\uparrow$ means higher metric values lead to better performance. **Bold** numbers highlight the best best performance.

| Dataset | Method | Structure validity$^\uparrow$ | COV-R$^\uparrow$ | COV-P$^\uparrow$ |
|---------|--------|------------------|--------|--------|
| Perov-5 | SyMat with coordinate score matching | **100.00%** | 97.46% | 95.32% |
| | SyMat with distance score matching | **100.00%** | **99.68%** | **98.64%** |
| Carbon-24 | SyMat with coordinate score matching | **100.00%** | 96.72% | 93.44% |
| | SyMat with distance score matching | **100.00%** | **100.00%** | **97.59%** |
| MP-20 | SyMat with coordinate score matching | **100.00%** | 94.25% | 95.61% |
| | SyMat with distance score matching | **100.00%** | **98.97%** | **99.97%** |

# E   Visualization Results

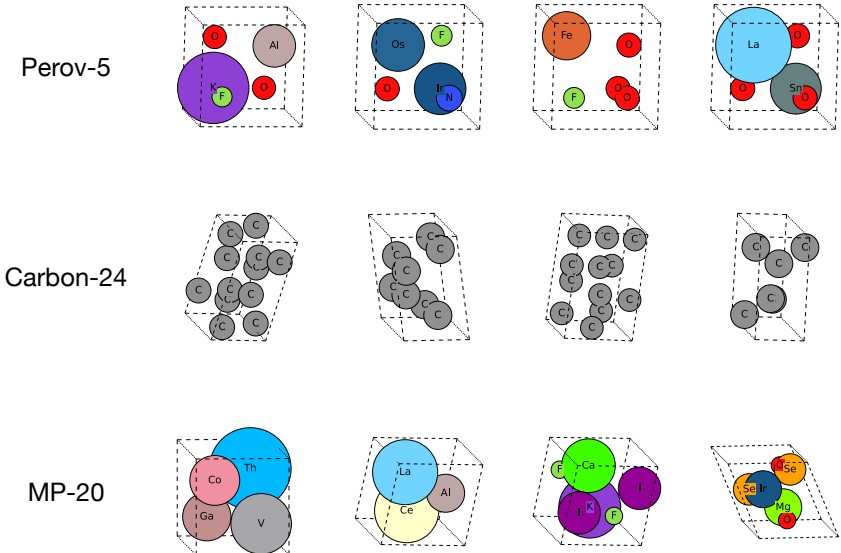

Figure 3: Illustrations of some periodic materials generated by SyMat models trained on Perov-5, Carbon-24, and MP-20 datasets.

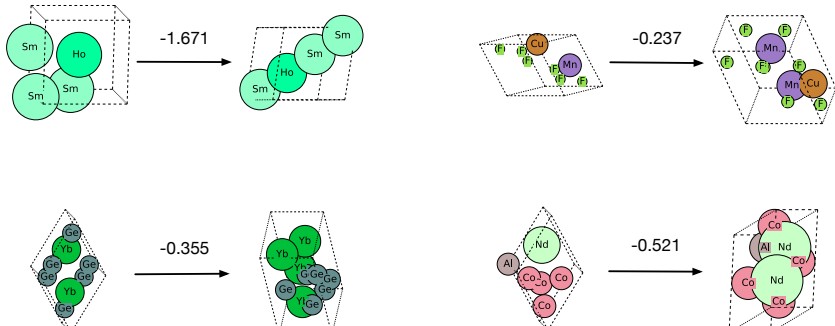

Figure 4: Illustrations of some periodic materials before and after being optimized by SyMat models in the property optimization task. Numbers show how energies (eV/atom) changes in optimization.

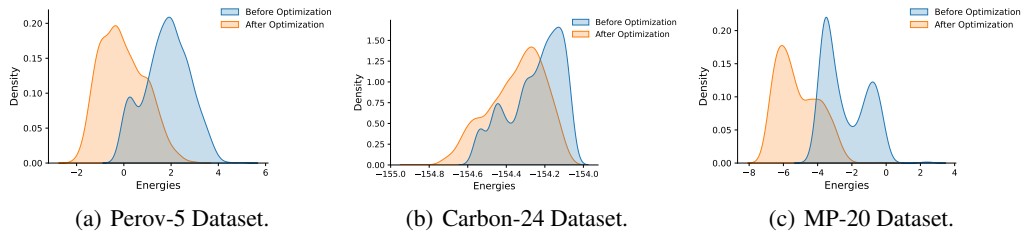

Figure 5: Illustrations of material energy (eV/atom) distribution before and after being optimized by SyMat models in the property optimization task.

