# OpenReview forum: "Towards Symmetry-Aware Generation of Periodic Materials"
_NeurIPS.cc/2023/Conference — NeurIPS 2023 spotlight_

### Official Review · Reviewer_ukbg · 2023-07-06

**Soundness:** 3 good
**Presentation:** 3 good
**Contribution:** 3 good
**Rating:** 7
**Confidence:** 3

**Summary:**

The problem of generating periodic materials with deep models is considered, focusing on capturing their unique symmetries. SyMat, a novel material generation approach, is proposed, using a variational auto-encoder to generate atom types, lattice lengths, and angles. A score-based diffusion model with a symmetry-aware probabilistic model is employed for generating atom coordinates, demonstrating promising performance in random generation and property optimization tasks.

**Strengths:**

- Clear presentation
- Good structure
- Fair comparison to prior work (CDVAE)
- Important application (materials discovery)
- Convincing empirical results
- Clear limitations section

**Weaknesses:**

- Contributions to ML method development minor

**Questions:**

- Eq. 2: Why did you decide to factorize $p(A, L)$ into $p(A) * p(L)$. I would have guessed that coupling $A$ and $L$ is important.
- Did you consider tackling the permutation symmetry in $[\ell, \phi]$, where permuting $\ell_1$ with $\ell_2$ leaves $\phi_{1,2}$ unaffected (or does it?).


**Limitations:**

As the authors comment, diffusion models are computationally intensive. I don't consider that a direct problem of the paper, though.

---

> ### Author Rebuttal · Authors · 2023-08-08
>
> Thank you for your positive feedback and comments! We hope your concerns and questions can be addressed by the following responses.
>
> > **Q1: Contributions to ML method development are minor.**
>
> We agree that our contributions to general generative models or other machine learning methods are not significant, as we focus on developing novel periodic material generation models in this work. Nonetheless, we make significant technical contributions in developing novel generative learning methods to achieve symmetry-aware generation of 3D periodic material structures. Specifically, for the purpose of achieving invariance to permutation, rotation, translation and periodic transformations, we propose a series of novel techniques in the probabilistic modeling of periodic material structures.
> They include transforming atom types and lattices to symmetry-invariant generation targets, and calculating coordinate score functions by the score functions of edge distances in multi-edge graphs. We believe these are highly non-trivial technical contributions, and our proposed novel model or techniques will potentially be very useful to developing automatic design systems for 3D periodic material structures or other 3D periodic point cloud structures.
>
> > **Q2: Eq. 2: Why did you decide to factorize $p(A,L)$ into $p(A) * p(L)$. I would have guessed that coupling $A$ and $L$ is important.**
>
> The factorization of $p(A,L)$ into $p(A) * p(L)$ comes from the observation that the distribution of lattices is approximately independent of atom types in real-world periodic materials. Practically, the synthesis of different periodic materials can have very different chemical reaction environments, e.g., different temperatures or pressures. Even if the atom type composition in reactants is fixed, under very different chemical reaction environments, the synthesized periodic materials will have very different 3D structures and a wide range of different lattice lengths and angles. Hence, lattices are often assumed to be independent of atom types in periodic materials and our method (and also previous periodic material generation method [1,2]) takes this assumption in probabilistic modeling. Actually, this assumption is reflected in our used datasets --- materials in Perov-5 dataset have very similar lattices but extremely different atom types, while materials in Carbon-24 dataset have very similar atom types but extremely different lattices.
>
> [1] Ren, Zekun, et al. "An invertible crystallographic representation for general inverse design of inorganic crystals with targeted properties." Matter 5.1 (2022): 314-335.
> [2] Xie, Tian, et al. "Crystal diffusion variational autoencoder for periodic material generation." ICLR 2022.
>
> > **Q3: Did you consider tackling the permutation symmetry in $[\ell, \phi]$, where permuting $\ell_1$ with $\ell_2$ leaves $\phi_{1,2}$ unaffected (or does it?)?**
>
> We did not consider the permutation symmetry in $[\ell, \phi]$. In data preprocessing, we always produce the three lattice lengths $\ell=[\ell_1,\ell_2,\ell_3]$ in an order satisfying $\ell_1\le\ell_2\le\ell_3$. We will clarify it in the revision of our paper.

---

### Official Review · Reviewer_sa6H · 2023-07-07

**Soundness:** 3 good
**Presentation:** 3 good
**Contribution:** 3 good
**Rating:** 5
**Confidence:** 3

**Summary:**

this paper proposes a method to enforce required symmetric properties in output generated by diffusion models. this method is applied to the problem of generation of new materials with desired symmetric properties. the method comprise of VAE for the generation of atom numbers, A, and lattice structures, L, while diffusion is used for the generation of coordinates, P. The main assumption in this work is that the probability of a molecule does not change under permutation/rotation+translation.

**Strengths:**

an insightful approach for incorporating required symmetric properties.
Figure 2 helps to understand the entire method.

**Weaknesses:**

the assumption on equiprobable molecules, P(M)=P(M'), is not true in general. Though it can work in some cases.

the assumption that "type vector A" is independent on "lattice matrix L" in the generation of a new material sounds heuristic. could you please elaborate of this assumption and its limitations?



**Questions:**

could you please explain why P(M) = P(M') in Section 3.1? There are inter-atom, A, interactions, which may affect the properties of a molecule, including its energy etc. The probability to find M' in nature can be significantly different from that of M.

So, I am not sure that P(M)=P(M') under 'Permutation' and/or 'Rotation and translation' of A and P is a reasonable assumption in general. If I am wrong, please explain. Otherwise, please explain the limitations of this assumption.

Could you please explain "Since the structures of atom types and lattices are relatively simple.."?
Is not it a combinatoric problem? What does it mean simple in this context? How is this simplicity related to VAE? VAE has quite a lot of drawbacks.

Why do you use different approaches for A+L and for P, VAE, and Diffusion, respectively? You mention that Diffusion is more powerful than VAE, so it can not 'harm' the quality of A+L in comparison to VAE?
Why to complicate?

line 187: "Afterwards, the decoder model uses zA and zL to reconstruct the exact $c$, $\ell$...". VAE usually has a reconstruction error. How do you guarantee "exact reconstruction"?

Can you explain why "our method cannot be applied to non-periodic materials"?

What are the possible directions for the acceleration of "atom coordinate generation"?




**Limitations:**

the assumption about the equiprobable molecule generation P(M) = P(M'), is limiting in my option.

the authors adequately address the limitations in Section 3.5.

---

> ### Author Rebuttal · Authors · 2023-08-08
>
> Thanks for your valuable feedback and comments! We make detailed responses below to address your concerns and questions. We hope you will raise your score if we address your concerns.
>
> > **Q1: Could you please explain why $P(M) = P(M')$ in Section 3.1? There are inter-atom, $A$, interactions, which may affect the properties of a molecule, including its energy etc. The probability to find $M'$ in nature can be significantly different from that of $M$. So, I am not sure that $P(M)=P(M')$ under 'Permutation' and/or 'Rotation and translation' of $A$ and $P$ is a reasonable assumption in general.**
>
> We think there is some misunderstanding. $M$ and $M'$ in Section 3.1 are not different periodic materials in nature, but actually different representations of the same periodic material in nature. Intuitively, for a certain periodic material in nature, we can write its atoms in any order, and think the 3D material structure as a rigid body and rotate or translate this rigid body arbitrarily in 3D space. These atom order permutation, rotation or translation do not change any internal structures or chemical properties of this material, but simply produce different equivalent representations of this material. In Section 3.1, $M$ and $M'$ are actually the equivalent representations of the same material produced in this way, so $P(M)=P(M')$.
>
> > **Q2: The assumption that "type vector $A$" is independent on "lattice matrix $L$" in the generation of a new material sounds heuristic. Could you please elaborate of this assumption and its limitations?**
>
> This assumption comes from the the observation that the distribution of lattices is approximately independent of atom types in real-world periodic materials. Practically, the synthesis of different periodic materials can have very different chemical reaction environments, e.g., different temperatures or pressures. Even if the atom type composition in reactants is fixed, under very different chemical reaction environments, the synthesized periodic materials will have very different 3D structures and a wide range of different lattice lengths and angles. Hence, lattices are often assumed to be independent of atom types in periodic materials and our method (and also previous periodic material generation method [1,2]) takes this assumption in probabilistic modeling. Actually, this assumption is reflected in our used datasets --- materials in Perov-5 dataset have very similar lattices but extremely different atom types, while materials in Carbon-24 dataset have very similar atom types but extremely different lattices. The potential limitations of this assumption are that in theory, generative models may accidently generate periodic materials whose lattice lengths are so small that the unit cell volume is not enough to cover all atoms in a unit cell (though we have not observed such phenomena in our experiments).
>
> [1] Ren, Zekun, et al. "An invertible crystallographic representation for general inverse design of inorganic crystals with targeted properties." Matter 5.1 (2022): 314-335.
> [2] Xie, Tian, et al. "Crystal diffusion variational autoencoder for periodic material generation." ICLR 2022.
>
> > **Q3: Could you please explain "Since the structures of atom types and lattices are relatively simple.."? Is not it a combinatorial problem? What does it mean simple in this context? How is this simplicity related to VAE? VAE has quite a lot of drawbacks.**
>
> The structures of atom types and lattices are simple because their symmetry-invariant representations (i.e., atom type sets, lattice lengths and angles) are low-dimensional vectors whose distributions are easy to be captured by VAE models. Generating atom types is not a combinatorial problem because we do not brute-forcely traverse all combinations, but sample from the distribution learned on datasets, whose atom types only occupy a small part of all combinatiions. We agree that VAE models have drawbacks and are not as powerful as diffusion models, but they are powerful enough for atom types and lattices and are fast in generation, so we use VAE models.
>
> > **Q4: Why do you use different approaches for $A$+$L$ and for $P$, VAE, and Diffusion, respectively? You mention that Diffusion is more powerful than VAE, so it can not 'harm' the quality of A+L in comparison to VAE? Why to complicate?**
>
> Diffusion models are more powerful than VAE models, but are slower than VAE models in generation. For the atom type $A$ and lattice $L$, their distributions are simple enough (see the response to Q3) to be captured by both VAE models and diffusion models. To achieve fast generation, we use VAE models. Such designs are also used by previous material generation methods [1,2].
>
> [1] Ren, Zekun, et al. "An invertible crystallographic representation for general inverse design of inorganic crystals with targeted properties." Matter 5.1 (2022): 314-335.
> [2] Xie, Tian, et al. "Crystal diffusion variational autoencoder for periodic material generation." ICLR 2022.
>
> > **Q5: Line 187: "Afterwards, the decoder model uses $z^A$ and $z^L$ to reconstruct the exact $c$, $\ell$". VAE usually has a reconstruction error. How do you guarantee "exact reconstruction"?**
>
> We agree that there is no guarantee for "exact reconstruction". What we mean here is that in the training of VAE models, we use the exact $c$, $\ell$ as the reconstruction target to calculate the reconstruction loss function. We will clarify it in the revision of our paper.
>
> > **Q6: Can you explain why "our method cannot be applied to non-periodic materials"?**
>
> Due to the space limitation in author rebuttal to each reviewer, we post the response to this question in author rebuttal to all reviewers.
>
> > **Q7: What are the possible directions for the acceleration of "atom coordinate generation"?**
>
> Due to the space limitation in author rebuttal to each reviewer, we post the response to this question in author rebuttal to all reviewers.

---

### Official Review · Reviewer_QiTe · 2023-07-07

**Soundness:** 3 good
**Presentation:** 3 good
**Contribution:** 3 good
**Rating:** 7
**Confidence:** 3

**Summary:**

This paper proposes SyMat, a new approach to generate materials that capture physical properties of perodic structures. This method generates atom types, lattice matrix of the unit cell, and then atom positions, all while maintaining invariance to permutations, rotation, translation, and periodic transformation. They use a VAE (with SphereNet as the encoder and MLPs as decoder) to generate the elements and lattice vectors, and score-based diffusion model to generate the atom positions. They evaluate SyMat in random generation and property optimization tasks.

**Strengths:**

- This paper tackles an important problem of generating periodic materials, useful for the field of materials discovery given that datasets of known materials for inference are limited
- Good techniques of how to frame the prediction task in a symmetry-aware way
- Good comparison to related work and including several of them as baselines
- Generally strong results, with SyMat performing best or second best in nearly all metrics

**Weaknesses:**

- I don't fully agree with not needing to evaluate the ability of the model to reconstruct materials from their latent representations. 1: even if the goal is to generate novel materials, we still want to ensure that the latent representation is valid and useful, and reconstruction of known materials seems to be a good way of validating the latent representations. 2: Agree that you can't use absolute coordinates for evaluation, but you should be able to evaluate it based on interatomic distances (or perhaps at least something like graph connectivity).
- Is there some way to evaluate diversity in the random samples generated? Also, ideally it would be great to see if the generated materials are stable (e.g. able to converge from some relaxation), but perhaps that's out of scope.

**Questions:**

- For success rate of property optimization, "the rate of decoded materials whose properties are in the top 5%, 10%, and 15% of the energy distribution in the dataset", what is "the dataset" referring to - is it just some energy threshold determined by the Nth percentile of the test set? Would it be helpful to report the distribution of energies genereated in addition to success rate at thresholds?
- How is the energy of the generated materials obtained? Is it verified by something like DFT? i.e. to make sure that the model isn't cheating by reporting artificially low energies

**Limitations:**

- The authors acknowledge limitations such as model inference speed and that the work is limited to task of periodic structures

---

> ### Author Rebuttal · Authors · 2023-08-08
>
> Thanks for your positive feedback and helpful suggestions! We believe your concerns and questions can be addressed by the following responses.
>
> > **Q1: I don't fully agree with not needing to evaluate the ability of the model to reconstruct materials from their latent representations. 1: even if the goal is to generate novel materials, we still want to ensure that the latent representation is valid and useful, and reconstruction of known materials seems to be a good way of validating the latent representations. 2: Agree that you can't use absolute coordinates for evaluation, but you should be able to evaluate it based on interatomic distances (or perhaps at least something like graph connectivity).**
>
> Thanks for your insightful comments and suggestions! We agree that material reconstruction experiments can help ensure that the latent representation is valid and useful. We have done the reconstruction experiment and evaluated the performance by the following metrics on the test sets of Perov-5, Carbon-24 and MP-20 datasets .
> - First, to show the validity and usefulness of latent representations $z^A$ and $z^L$, we evaluate the percentage of materials whose atom types can be exactly reconstructed (Atom Type Match Rate), and the root mean square error in lattice lengths and angles between the target and reconstructed materials (Lattice RMSE).
> - Second, to show the score-based diffusion model has the capacity to approximately reconstruct 3D structures, we evaluate the root mean square error in interatomic distances between the target and reconstructed materials (Distance RMSE).
>
> We present the experimental results of SyMat together with baseline methods on Perov-5, Carbon-24 and MP-20 datasets in the following three tables.
>
> |Method|Atom Type Match Rate|Lattice RMSE|Distance RMSE|
> |----|----|----|----|
> |FTCP|**99.67%**|0.0786|0.2953|
> |Cond-DFC-VAE|58.92%|0.0765|0.2281|
> |CDVAE|98.16%|0.0231|0.1072|
> |SyMat|98.3%|**0.0224**|**0.0723**|
>
> |Method|Atom Type Match Rate|Lattice RMSE|Distance RMSE|
> |----|----|----|----|
> |FTCP|**100%**|0.1349|0.3759|
> |CDVAE|**100%**|**0.0624**|0.1745|
> |SyMat|**100%**|0.0632|**0.1186**|
>
> |Method|Atom Type Match Rate|Lattice RMSE|Distance RMSE|
> |----|----|----|----|
> |FTCP|71.46%|0.1057|0.2062|
> |CDVAE|68,32%|0.0532|0.0975|
> |SyMat|**72.1%**|**0.0510**|**0.0876**|
>
> Note that Cond-DFC-VAE can only be used to Perov-5 dataset in which materials have cubic structures, and G-SchNet and P-SchNet do not produce latent representations so they cannot do reconstruction. Results show that our method performs well in atom types, lattices and interatomic distances reconstruction.
>
> > **Q2: Is there some way to evaluate diversity in the random samples generated? Also, ideally it would be great to see if the generated materials are stable (e.g. able to converge from some relaxation), but perhaps that's out of scope.**
>
> - In our opinion, we can evaluate diversity of the randomly generated atom types by uniqueness percentage, which is the percentage of unique atom types among all the generated atom types. However, we think there is no way to evaluate the diversity in 3D structures. We present the uniqueness percentages of SyMat and baseline methods on Perov-5 and MP-20 datasets (Carbon-24 is not used because all materials in Carbon-24 are composed of only carbon atoms, so the uniqueness is expected to be low) in the following table.
>
> |Method|Perov-5|MP-20|
> |----|----|----|
> |FTCP|72.33%|79.04%|
> |Cond-DFC-VAE|80.36%|---|
> |G-SchNet|98.74%|99.23%|
> |P-G-SchNet|98.57%|99.03%|
> |CDVAE|98.61%|99.84%|
> |SyMat|**99.43%**|**99.98%**|
>
> Note that Cond-DFC-VAE can only be used to Perov-5 dataset in which materials have cubic structures. Results show that our method performs well in generating materials with diverse atom types.
>
> - We agree that it would be practically meaningful to evaluate if the generated materials are stable. However, it needs very expensive computational resources to do the structure relaxation, so we and existing studies [1,2] in material generation evaluate the stability of generated materials by structure validity, i.e., the percentage of generated materials whose interatomic distances are all larger than 0.5Å. We think the evaluation by structure relaxation is out of scope and leave it to the future work.
>
> [1] Court, Callum J., et al. "3-D inorganic crystal structure generation and property prediction via representation learning." Journal of Chemical Information and Modeling 60.10 (2020): 4518-4535.
> [2] Xie, Tian, et al. "Crystal diffusion variational autoencoder for periodic material generation." ICLR 2022.
>
> > **Q3: For success rate of property optimization, "the rate of decoded materials whose properties are in the top 5%, 10%, and 15% of the energy distribution in the dataset", what is "the dataset" referring to - is it just some energy threshold determined by the Nth percentile of the test set? Would it be helpful to report the distribution of energies generated in addition to success rate at thresholds?**
>
> Yes, success rates are computed based on the energy threshold determined by the 5, 10, and 15 percentile of the test set. We agree that it is helpful to report the distribution of energies in addition to success rate. We have visualized the energy distribution before and after property optimization in Figure 3 of the rebuttal PDF attached in author rebuttal to all reviewers.
>
> > **Q4: How is the energy of the generated materials obtained? Is it verified by something like DFT? i.e. to make sure that the model isn't cheating by reporting artificially low energies.**
>
> To ensure fair comparison, we follow the setting in CDVAE [1] to obtain energies by predicting them from a GNN model provided in CDVAE official codes. From our observation, this GNN model does not report unusually low energies.
>
> [1] Xie, Tian, et al. "Crystal diffusion variational autoencoder for periodic material generation." ICLR 2022.

---

> > ### Comment · Reviewer_QiTe · 2023-08-20
> >
> > Thank you for your response and for providing the additional results. I maintain my recommendation for acceptance for this paper.

---

> > > ### Author Response · Authors · 2023-08-20
> > > **Response**
> > >
> > > Thank you for your response! We are glad to know that your will maintain your acceptance decision. Thank you again for your positive feedback and valuable suggestions!

---

### Official Review · Reviewer_oTTM · 2023-07-19

**Soundness:** 3 good
**Presentation:** 3 good
**Contribution:** 3 good
**Rating:** 7
**Confidence:** 1

**Summary:**

This paper introduces a deep learning based method for the automatic generation of periodic molecules. At the core of their method lies a variational autoencoder and a score-based diffusion models, which generate lattice lengths and angles, and atom coordinates, respectively. Theoretical derivations show that their method is invariant to symmetric transformations and quantitative validation show that the proposed method can be used for optimization problems.

I have read the rebuttal. I am more positive about this submission after the rebuttal and reading the other reviews. The authors have addressed my concerns regarding empirical analysis of limitations, and the quality of writing and figures.

**Strengths:**

- The paper is overall well written and easy to follow.
- Extensive validation is provided, with comparisons with previous work across a variety of metrics and with ablation studies of the components of their own method.
- Invariance to symmetric transformations is achieved through theoretical derivations, which makes the model more robust to these transformations than data augmentation-based methods.
- The provided supplementary material is quite solid and shows very interesting results and ablations.
- The paper and supplementary material contain enough detail to help reproducibility of the results.

**Weaknesses:**

- I believe the paper could do a better job at motivating the need for modelling symmetry in molecule generation. I am no expert in this topic and I lacked some motivation onto why the problem is salient.
- The figures in the paper could be improved, particularly Figure 2 could benefit from better illustrations.
- Some design choices could be motivated better, particularly related to the model design.
- The paper needs, in my opinion, more qualitative results like those provided in Figure 3 in the supplementary material.
- The potential limitations of SyMat are not thorougly analyzed.

**Questions:**

- Will code be provided?
- Why is denoising score matching "more effective to train" the generative model?

**Limitations:**

In my opinion, the authors could provide more discussion of the limitations of their work.

---

> ### Author Rebuttal · Authors · 2023-08-08
>
> Thank you for your positive feedback and insightful comments! We hope your concerns and questions can be addressed by the following responses.
>
> > **Q1: The paper could do a better job at motivating the need for modeling symmetry.**
>
> Thanks for your suggestion. The main motivation of modeling symmetry comes from property optimization. A significant target of developing generative models for periodic materials is to generate novel periodic materials with desirable chemical properties (e.g., low energies). Notably, chemical properties are invariant to physical symmetry transformations. For instance, rotating a material structure in 3D space does not change its energy. In other words, chemical properties only depend on the 3D geometric information that is invariant to symmetry transformations (e.g., interatomic distances). Using symmetry-aware modeling forces generative models to only capture the distribution of these 3D geometric information, which facilitates searching materials with desirable properties in the latent space. We will add the above discussions to the revision of our paper.
>
> > **Q2: The figures in the paper could be improved, particularly Figure 2 could benefit from better illustrations.**
>
> Thanks for your suggestion. We have improved Figure 2 by adding intuitive graphics instead of only using mathematical symbols to represent the generation targets (see Figure 1 in the rebuttal PDF attached in author rebuttal to all reviewers). We will update Figure 2 in the revision of our paper.
>
> > **Q3: Some design choices could be motivated better, particularly related to the model design.**
>
> Thanks for your suggestion. Due to the space limitation in author rebuttal to each reviewer, we post the response to this question in author rebuttal to all reviewers.
>
> > **Q4: The paper needs more qualitative results like those provided in Figure 3.**
>
> Thanks for your suggestion. We have included more qualitative visualization results of several optimized periodic materials obtained in the property optimization task in Figure 2 of the rebuttal PDF attached in author rebuttal to all reviewers. We will add them to the revision of our paper.
>
> > **Q5: The potential limitations of SyMat are not thoroughly analyzed.**
>
> Thanks for your suggestion. We thoroughly analyze the potential limitations of SyMat below.
> - First, SyMat takes as long as an hour to generate periodic materials in one time, and most of the time is spent on atom coordinate generation. This is because generating atom coordinates requires to run thousands of Langevin dynamics sampling steps. To tackle this limitation, we will explore using stochastic differential equation (SDE) based diffusion models [1,2] for atom coordinate generation in the future as they need much less sampling steps.
> - Second, SyMat is designed to maintain invariance to periodic transformations, so it cannot be applied to non-periodic materials as this will lead to incorrect probabilistic modeling. Though periodic materials occupy a large part of materials used in real-world applications, this limitation hampers the application to some useful non-periodic materials, such as amorphous solids and polymers. In the future, we will explore developping generative models that can capture symmetry properties in non-periodic materials.
> - Third, our work follows previous material generation studies [3] to evaluate the quality of the generated materials by some basic metrics, such as composition and structure validity. However, they do not use metrics to evaluate the synthesizability of the generated materials, though these metrics are useful for practical applications. In the future, we will collaborate with experts in material science to come up with metrics for evaluating synthesizability of the materials generated by our method.
>
> We will add the above discussions to the revision of our paper.
>
> [1] Song, Yang, et al. "Score-based generative modeling through stochastic differential equations." ICLR 2021.
> [2] Kingma, Diederik, et al. "Variational diffusion models." NeurIPS 2021.
> [3] Xie, Tian, et al. "Crystal diffusion variational autoencoder for periodic material generation." ICLR 2022.
>
> > **Q6: Will code be provided?**
>
> Yes, we will release our code after the paper decision is out.
>
> > **Q7: Why is denoising score matching "more effective to train" the generative model?**
>
> Denoising score matching can effectively train the model because it significantly improves the prediction accuracy of scores in low data density regions. In most practical application scenarios, the data whose distribution will be captured by generative models is not universally scattered in the whole data space, but concentrated in one or several small regions. Hence, there exist large amounts of low data density regions where the probabilistic density of data distribution is low. If the model is only trained to accurately predict scores of data samples in the given dataset, it will not perform well in predicting scores in low data density regions. However, denoising score matching first moves dataset samples out from high data density regions by adding Gaussian noise to them, then the model is trained to predict the score of the noised data sample. This mechanism forces the model to achieve high score prediction accuracy in low data density regions. Since the data generation process of models (Langevin dynamics sampling) needs accurate scores in the whole data space, the model trained by denoising score matching will generate high-quality data in more chances. We recommend Section 3 of [1] as an excellent introduction about this topic.

---

> > ### Comment · Reviewer_oTTM · 2023-08-16
> > **Response to rebuttal**
> >
> > I am glad my comments were useful! Thank you for your detailed response. I am also glad that code will be shared
> >
> > I suggest you include a summary of your excellent responses to my Q3 and Q7 on the paper, as it should help motivate the design choices, which will increase the potential impact of the paper.

---

> > > ### Author Response · Authors · 2023-08-16
> > > **Response**
> > >
> > > Thank you for your feedback and suggestion! We are glad that you find our responses helpful. We will definitely add a summary of our responses to Q3 and Q7 to the revision of our paper.

---

### Author Rebuttal · Authors · 2023-08-08

We sincerely appreciate all reviewers' hard work in reviewing our paper and writing comments. We are glad that all reviewers give positive feedbacks to our work and make many insightful suggestions to help improve our paper.

In the rebuttal PDF file, we include an updated version of the original Figure 2 in our paper following **reviewer oTTM**'s suggestion, visualize several optimized periodic materials obtained in the property optimization task following **reviewer oTTM**'s suggestion and visualize the energy distribution before and after property optimization following **reviewer QiTe**'s suggestion.

We provide detailed responses to resolve each reviewer's questions or concerns in the author rebuttal to each reviewer. As the space of author rebuttal to each reviewer is limited, we have included some additional responses here.

Additional responses to **reviewer oTTM**:

> **Q3: Some design choices could be motivated better, particularly related to the model design.**

Thanks for your suggestion. We elaborate the motivation of our model design choices below.
- First, the structures of atom types and lattices are relatively simple and their symmetry-invariant representations (i.e., atom type sets, lattice lengths and angles) are low-dimensional vectors. Their distributions are simple enough to be captured by VAE models [1]. Also, VAE models are faster than diffusion models in generation speed. Hence, we use VAE models for atom type and lattice generation.
- Second, the structure of atom coordinates is much more complicated than atom types or lattices, so we use more powerful score-based diffusion models [2] to generate them. More importantly, using score-based diffusion models for atom coordinates enables us to trickly convert the prediction of coordinate scores to that of interatomic distance scores by the chain rule of derivatives. This strategy maintains invariance to rotation, translation and periodic transformations. We think it is hard to develop such a symmetry-aware probabilistic modeling for other generative models. Hence, we use score-based diffusion models for coordinate generation.
- Third, SphereNet [3] is chosen as the backbone network of our model. SphereNet is a 3D graph neural network for 3D graph representation learning. It has powerful capacities in 3D structure feature extraction because it considers complete 3D geometric information contained in interatomic distances, line angles and plane angles (torsion angles). Also, the features extracted by SphereNet are invariant to rotation, translation and periodic transformations when input 3D graphs are created by multi-graph method [4]. Because of its powerful capacity and symmetry-aware feature extraction, we use SphereNet as the backbone network.

We will add the above discussions to the revision of our paper.

[1] Kingma, Diederik P., and Max Welling. "Auto-encoding variational bayes." ICLR 2014.
[2] Song, Yang, and Stefano Ermon. "Generative modeling by estimating gradients of the data distribution." NeurIPS 2019.
[3] Liu, Yi, et al. "Spherical message passing for 3D molecular graphs." ICLR 2022.
[4] Xie, Tian, and Jeffrey C. Grossman. "Crystal graph convolutional neural networks for an accurate and interpretable prediction of material properties." Physical review letters 120.14 (2018): 145301.

Additional responses to **reviewer sa6H**:

> **Q6: Can you explain why "our method cannot be applied to non-periodic materials"?**

Our method is designed to maintain invariance to periodic transformations, but non-periodic materials are not invariant to periodic transformations, so applying our method to non-periodic materials leads to incorrect probabilistic modeling.

> **Q7: What are the possible directions for the acceleration of "atom coordinate generation"?**

We think possible and promising directions for acceleration are using stochastic differential equation (SDE) based diffusion models [1,2] because they need significantly less sampling steps in generation. We will explore these directions in the future.

[1] Song, Yang, et al. "Score-based generative modeling through stochastic differential equations." ICLR 2021.
[2] Kingma, Diederik, et al. "Variational diffusion models." NeurIPS 2021.

---

### Decision · Program_Chairs · 2023-09-21

**Decision:**

Accept (spotlight)

**Comment:**

The paper received positive ratings. The reviewers in general appreciate the paper’s effort to address an important problem of generating periodic materials, it's clear presentation, and the verification of the model through convincing and extensive results. The ACs agree with the reviewers and recommend the acceptance. The final version should, where possible, include a more detailed description of the model design as suggested by the reviewers.